# Constructing the human brain metabolic connectome with MR spectroscopic imaging reveals cerebral biochemical organization

Federico Lucchetti [1], Edgar Céléreau [1], Pascal Steullet [1], Yasser Alemán-Gómez [2], Patric Hagmann[2], Antoine Klauser [3] & Paul Klauser [1,4] ✉

Network science has mapped brain structural and functional organization, yet capturing metabolic contrast remains a major gap in connectomics. Using fast, high-resolution 3D whole-brain proton magnetic resonance spectroscopic imaging ([1]H-MRSI), we derive a within-subject metabolic connectome in 51 healthy subjects, defined as pairwise correlations among five metabolites (tCr, tNAA, Glx, Ins, Cho) across gray-matter parcels. Results show stability, consistency and replicability, including validation in an independent sample ($N = 13$) scanned at a different site. A dimensionality reduction analysis shows that the leading metabolic similarity mode forms a continuous caudal-to-rostral gradient across gray-matter regions. We show that this progression is reflective of a balance between local metabolic homogeneity and global metabolic diversity, and can be summarized by a principal path through the metabolic network. While the most metabolically active regions overlap with structural hubs, overall metabolic similarity aligns weakly with tractography-based structural connectivity but more closely with cytoarchitectonic similarity and gene co-expression matrices. These findings introduce the metabolic similarity gradient as a signature of the brain's biochemical organization and position MRSI as a biologically grounded dimension for connectomics in health and disease.

Since their inception in the 1980s, magnetic resonance imaging (MRI) techniques have become central to both clinical and basic neuroscience. MRI offers non-invasive, non-ionizing measurements of the living brain across various ages, with relatively quick acquisition times and moderate costs, making it widely accessible. MRI techniques have not only enhanced our understanding of the brain's anatomical structure, composition, and function but also enabled the construction of large-scale brain network models. In particular, diffusion MRI maps water diffusion along neuronal axons, revealing white matter tracts and pioneering the structural connectome and the field of connectomics[1,2]. BOLD (blood-oxygen-level-dependent) functional MRI introduces a temporal dimension to imaging, enabling functional connectivity to be defined as statistical dependencies–typically correlations–between regional BOLD time series, which index hemodynamic changes used as a proxy for synchronized neural activity[3]. The field of connectomics has since revealed that, like many other complex systems, both structural and functional brain networks exhibit fundamental organizational principles. The most prominent example is

[1]Center for Psychiatric Neuroscience, Department of Psychiatry, Lausanne University Hospital and University of Lausanne, Lausanne, Switzerland. [2]Connectomics Lab, Department of Radiology, Lausanne University Hospital and University of Lausanne, Lausanne, Switzerland. [3]Swiss Innovation Hub, Siemens Healthineers International AG, Lausanne, Switzerland. [4]Division of Child and Adolescent Psychiatry, Department of Psychiatry, Lausanne University Hospital and University of Lausanne, Lausanne, Switzerland. ✉e-mail: paul.klauser@unil.ch

modular architecture, which reflects a balance between local specialization and global integration[4–6]. This trade-off is a hallmark of healthy brain organization and provides a framework for understanding how alterations in network structure relate to behavior and disease. Individual differences in network organization are linked to cognition and personality[7], and their disruptions, are increasingly recognized as central to neurological and psychiatric disorders[8–12]. Within this framework, network abnormalities often converge on inter-hub connector nodes, which incur high metabolic costs through oxidative glucose metabolism[13] and are vulnerable to homeostatic disturbances. These hubs thus represent potential points of failure across diverse pathological conditions[6,11]. Diffusion MRI and tractography can capture macro-scale changes in the structural connectome, but remain limited in detecting microcircuit-level alterations[14]. Functional MRI and dynamic causal modeling provide complementary insights into metabolic activity[15–17], yet are constrained by computational demands, model specificity, and interpretative complexity[18]. These neuroimaging modalities, along with the network architectures they help reveal, have excelled at characterizing brain connectivity at the macro- to meso-scale. However, they all fundamentally rely on indirect proxies for assessing underlying cellular networks which are deeply shaped and constrained by the brain's underlying metabolic organization. Therefore, conventional MRI-based techniques have left a significant gap in connectomics research and limited our ability to link observed changes to their neurobiological basis. While ongoing advances in hardware, pulse sequences, and accelerated acquisition hold promise for pushing the spatial resolution of MRI toward finer scales, such improvements alone are unlikely to close this explanatory gap. Ultimately, what remains missing is not simply a matter of spatio-temporal resolution, but of accessing an additional dimension of information which offers more direct insights into the biochemical and physiological processes.

Once the dominant water and lipid signals are suppressed, magnetic resonance spectroscopy (MRS) reveals this hidden dimension— the chemical shift—which enables the separation and quantification of brain metabolites by their unique resonance frequencies. Building on this principle, recent advances in proton magnetic resonance spectroscopic imaging ($^1$H-MRSI)[19–26] now allow for whole-brain 3D metabolite mapping at high spatial resolution with substantially reduced acquisition times ( ~ 20 minutes for the 3D compressed-sensing SENSE low-rank $^1$H FID-MRSI sequence used in this study). Detailed and scalable analyses of brain metabolism are now facilitated and can be seamlessly integrated into clinical or research MRI protocols. At a magnetic field strength of 3T, the spectrally resolved metabolites include N-acetylaspartate + N-acetylaspartylglutamate (tNAA), creatine + phosphocreatine (tCr), glutamate + glutamine (Glx), phosphocholine + glycerophosphocholine (Cho), and myo-inositol (Ins). Specifically, tNAA may reflect neuronal integrity, tCr is a marker of energy metabolism, Glx is involved in excitatory neurotransmission, Cho is linked to membrane turnover, and Ins is associated with glial activity[27].

Given that these metabolites are critical for normal cellular functioning, and in light of previous MRS studies showing how their interplay impacts various neurological and psychiatric disorders[28–31], it is logical to suppose that these five key compounds must be delicately regulated by homeostatic processes[32] to maintain healthy brain function; hence, they participate in a broader neurometabolic network[33]. In this work, we propose to leverag high-resolution 3D $^1$H-MRSI to construct, a large-scale, within-subject MRI-based metabolic connectome, by mapping the regional similarity among these five metabolites across the gray matter of the entire brain.

We explore the feasibility of constructing an MRSI metabolic connectome via an individual metabolic similarity matrix (MetSiM) using data from a single whole-brain scan $^1$H-MRSI in two cohorts of healthy participants. We test five central hypotheses regarding its network structure: (1) its topological characteristics mirror those of naturally occurring networks; it correlates with (2) structural connectivity, (3) cytoarchitectonic similarity, and (4) genetic co-expression patterns; and (5) regions with the highest structural connectivity coincide with those of high metabolic levels.

## Results

### Metabolic similarity matrices construction

To evaluate the feasibility of a within-subject metabolic similarity matrix (MetSiM), we analyzed $^1$H-MRSI data from 51 healthy adolescents in Geneva (i.e., Mindfulteen study[34]) following the similarity matrix construction pipeline schematically laid out in Fig. 1. After partial-volume and MRI point spread function correction (see Methods MRSI Partial Volume Correction; representative MRSI shown in Supplementary Fig. 9), each subject's T1-weighted volume was parcellated using the Chimera LFMIHIFS-3 atlas. This yielded 277 cortical, subcortical, and cerebellar GM regions (see Methods Anatomical), which were coregistered to the individual MRSI space and directly applied to the MRSI data, thereby minimizing voxel interpolation and smoothing artifacts. Following an approach similar to that used for extracting morphometric profiles[35] (see Methods Estimation of Metabolic Profiles), we derived metabolic profiles from five spectrally resolved $^1$H-MRSI compounds (tNAA, tCr, Glx, Cho, Ins) and retained 210 $z$-score-normalized profiles corresponding to brain regions with reliable $^1$H-MRSI coverage (see Supplementary Information Fig. 1).

The metabolic profiles were augmented via a Monte Carlo-driven uncertainty propagation procedure (see Methods Estimation of Metabolic Profiles), in which each measurement was perturbed $K_{pert}$ times based on its estimated variance (Cramér-Rao lower bounds issued by the LCModel metabolite quantification maps, shown in Supplementary Fig. 10), which effectively increased sample size per metabolite signal by a factor of $K_{pert}$.

Pairwise Spearman correlations between regional metabolic profiles were calculated to create a 210 × 210 MetSiM for each individual. Figure 2a shows an example MetSiM for $K_{pert}$ = 50, highlighting strong positive correlations clustering along the diagonal, representing super-regional communities (cortex, subcortex, cerebellum). Symmetrical clusters between hemispheres indicate metabolic similarity in corresponding regions, while negative correlations off-diagonally, particularly between distinct super-regions (e.g., cortex and subcortex), reflect metabolic dissimilarity.

### Stability, consistency and replicability

We evaluated the stability of MetSiM construction by testing two key parameters: the contribution of individual metabolites and the $K_{pert}$ values. A leave-one-metabolite-out approach assessed the influence of individual metabolites by comparing the original 5-metabolite MetSiM with MetSiM matrices reconstructed after imputing a single metabolite. This was repeated for three $K_{pert}$ values (1, 50, 100) in all individuals. For each test, pairwise edge strengths from the original and imputed matrices were correlated for each individual's MetSiM, and these correlations were aggregated across all subjects using Fisher's z-transformation, with $p$-values corrected using the Benjamini–Hochberg false discovery rate method. At $K_{pert}$ = 1, correlations between the original MetSiM and those reconstructed with a single metabolite omitted were low ($r$ = 0.32–0.40, $p$ < 0.05). Increasing $K_{pert}$ to 50 substantially improved these correlations (tNAA: $r$ = 0.80, Ins: $r$ = 0.96, Cho: $r$ = 0.81, Glx: $r$ = 0.88, tCr: $r$ = 0.96; all $p$ < 0.05), with only marginal gains at $K_{pert}$ = 100 ($r$ = 0.81–0.97, $p$ < 0.05). To further examine differences in metabolite contributions, an ANOVA on the $K_{pert}$ = 50 correlations revealed a significant effect ($F$ = 1154, $p$ < 0.001), and post hoc Tukey HSD tests indicated that tNAA, Cho, and Glx contributed significantly more to MetSiM construction than Ins or tCr, whose removal had minimal impact on correlations.

# Metabolic Similarity Matrix Construction

**Fig. 1 | Within-subject metabolic similarity matrix construction. a–b** Acquisition of ¹H-MRSI and anatomical T1-weighted images, followed by the reconstruction of 5 spectrally resolved metabolite brain signals, resulting in 5 volume maps ([*tNAA, tCr, Glx, Cho, Ins*]. Metabolite levels shown in institutional units [I.U.]. **c** Anatomical volume parcellation using the Chimera parcellation scheme[77], producing $n$ brain parcels. **d** Coregistration of the anatomical image to the ¹H-MRSI space and mapping of the five metabolite signals to the $n$ brain parcels. **e** Calculation of the median metabolite signal per parcel, followed by z-score normalization across all brain parcels per metabolite for each individual, to generate standardized and z-score normalized metabolic profiles $\{\mathbf{z}_m\}_{m=1,\ldots,n}$ at region $i$. **f** Pairwise Spearman correlations between metabolic profiles across all brain regions, yielding a within-subject metabolic similarity matrix (MetSiM).

The consistency of MetSiMs was evaluated by comparing individual MetSiMs with a group-level MetSiM, derived by averaging all individual MetSiMs from the Geneva study (Fig. 2b). Variability was quantified by calculating edge-wise correlations between individual- and group-level MetSiMs, Fisher-transformed, and aggregated for overall alignment. Correlations increased from $r = 0.51$ at $K_{pert} = 1$ to $r = 0.60$ at $K_{pert} = 50$, with a minor improvement at $K_{pert} = 100$ ($r = 0.61$), showing that $K_{pert} = 50$ sufficiently reduces variability and ensures consistent MetSiM construction. We fixed $K_{pert} = 50$ for subsequent analyses. Full results are reported in Supplementary Table 1. Replicability was assessed using a different sample of healthy participants from the Lausanne Psychosis Cohort[36] scanned on a different MRI platform. The group-level MetSiMs for the Geneva and Lausanne studies (shown on Fig. 2c) were strongly correlated ($r = 0.86$, $p < 0.001$), demonstrating the replicability of the MetSiM construction method across sites (i.e., different sample and scanner).

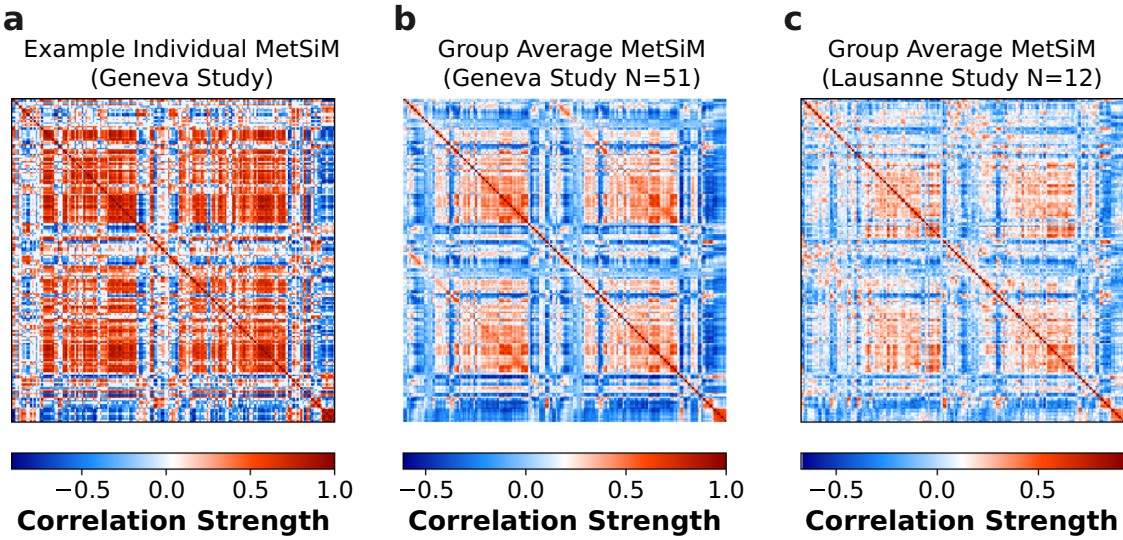

**Fig. 2 | Individual and Group Average Metabolic Similarity Matrices. a** Individual MetSiM ($N = 1$), from the Geneva study **b** Group averaged ($N = 51$) MetSiM from the Geneva study. **c** Group averaged ($N = 13$) MetSiM from the Lausanne study.

## Metabolic similarity mode and gradients

To show the spatial organization of metabolic similarity and the spatial discrimination between positive and negative MetSiM weights, we analyzed pairwise group-averaged MetSiM weights ($N = 51$, Geneva study) in relation to inter-regional Euclidean distances, pooling data from both hemispheres (Fig. 3a).

We then binned positive and negative MetSiM values separately in 2 mm intervals, generating density distributions (Fig. 3b). Positive MetSiM values cluster at shorter distance ranges (37 mm), whereas negative MetSiM values peak at longer distance ranges (71 mm) indicating a clear inverse spatial dependence. This pattern suggests that metabolic similarity reflects a spatial gradient of local metabolic coupling and distal metabolic differentiation.

To clearly visualize its spatial embedding, we reduced the 210 × 210 group-averaged MetSiM matrix to a 210 × 1 feature vector using Principal Component Analysis (PCA) for denoising and variance capture, followed by t-distributed Stochastic Neighbor Embedding (t-SNE) to preserve the observed local connectivity patterns in the MetSiM matrix (see Methods Metabolic Similarity Mode; first component accounts for 53% of variance). The resulting 210 × 1 vector–the first Metabolic Similarity Mode (MS mode)–was mapped back onto the anatomical locations of the corresponding brain parcels in the Montreal Neurological Institute (MNI) standard stereotaxic space, enabling 3D visualization of metabolic similarity patterns (Fig. 3c). The MS mode was inverse-mapped onto the five z-score-normalized metabolite level profiles (Fig. 3c; with color scale), allowing each MS mode value to be interpreted as a reduced representation of the five-feature metabolic profile. Common metabolite ratios were mapped onto the MS mode (Supplementary Fig. 11) and are consistent with previously reported regional MRSI ratio patterns. MS mode map generation is robust to cortical parcellation (Supplementary Fig. 3), showing high alignment in terms of normalized mutual information (MI) with alternative schemes: Schaefer197 (MI = 0.65), MIST197 (MI = 0.64), 15 × 15 × 15 mm grid (MI = 0.99), and 10 × 10 × 10 mm grid (MI = 0.99).

This representation reveals two distinct yet co-occurring features: first, a local property characterized by a high degree of local similarity in MS mode values; and second, a global organizational pattern defined by a high degree of diversity in MS mode values across regions.

The first property could result from spatial blurring inherent in MRI acquisition or from the underlying biological organization.

Regardless of its origin, we tested to what extent the observed MetSiM structure could be accounted for by spatial autocorrelation alone, using a distance-based random geometric null model (RandGeom) (see Methods Spatial Null Models) which reproduces both the distance dependence and the heterogeneity of variance observed in the empirical MetSiM distribution (Fig. 3b).

We generated 1000 RandGeom models and correlated each with the empirical MetSiM, producing a distribution of correlation coefficients (Supplementary Fig. 4 and Supplementary Table 2). The best RandGeom model (top, Fig. 3d) achieved only $r = 0.082$ and failed to reproduce key features such as bilateral homotopy or separation of nearby but non-adjacent regions (e.g., cerebellum-occipital cortex), as shown by the MS mode map of the simulated MetSiM (right, Fig. 3d). Enforcing hemispheric symmetry - by fitting to one hemisphere and mirroring across the midline - improved correlations (LH: $r = 0.114$, RH: $r = 0.131$). Further constraining connections to gray matter-adjacent regions (RandGeom · GMAdj) produced the highest correlations (LH +RH: $r = 0.133$, LH: $r = 0.208$, RH: $r = 0.223$), capturing local adjacency patterns but still lacking full bilateral symmetry in the whole brain (LH +RH) model (bottom, Fig. 3d). These results show that spatial proximity alone fails to reproduce key features of the MetSiM structure, such as bilateral homotopy and the separation of nearby but distinct regions. Adding biologically grounded constraints (gray matter adjacency and hemispheric symmetry) improves alignment with the empirical data, indicating that metabolic similarity cannot be reduced to spatial autocorrelation from geometric distance or imaging-induced smoothness, but instead reflects underlying biological organization at the systems level.

The second property corresponds to a global organizational pattern, with the MS mode defining the principal axis of variation in the MetSiM and producing a continuous, widely distributed range of values across the brain. This axis separates cortical from subcortical regions, differentiates within the cortex along a rostro-caudal trajectory, and distinguishes neighboring but functionally distinct areas such as the primary motor cortex (precentral gyrus) and primary somatosensory cortex (postcentral gyrus). The limbic (e.g., cingulate and parahippocampal cortex, thalamus, hippocampus) and paralimbic (e.g., insula) systems, along with related subcortical nuclei (e.g., striatum), exhibit metabolic profiles clearly distinct from the isocortex. Notably, rostro-caudal progression in the MS mode variation is also evident within individual structures such as the cingulate cortex and

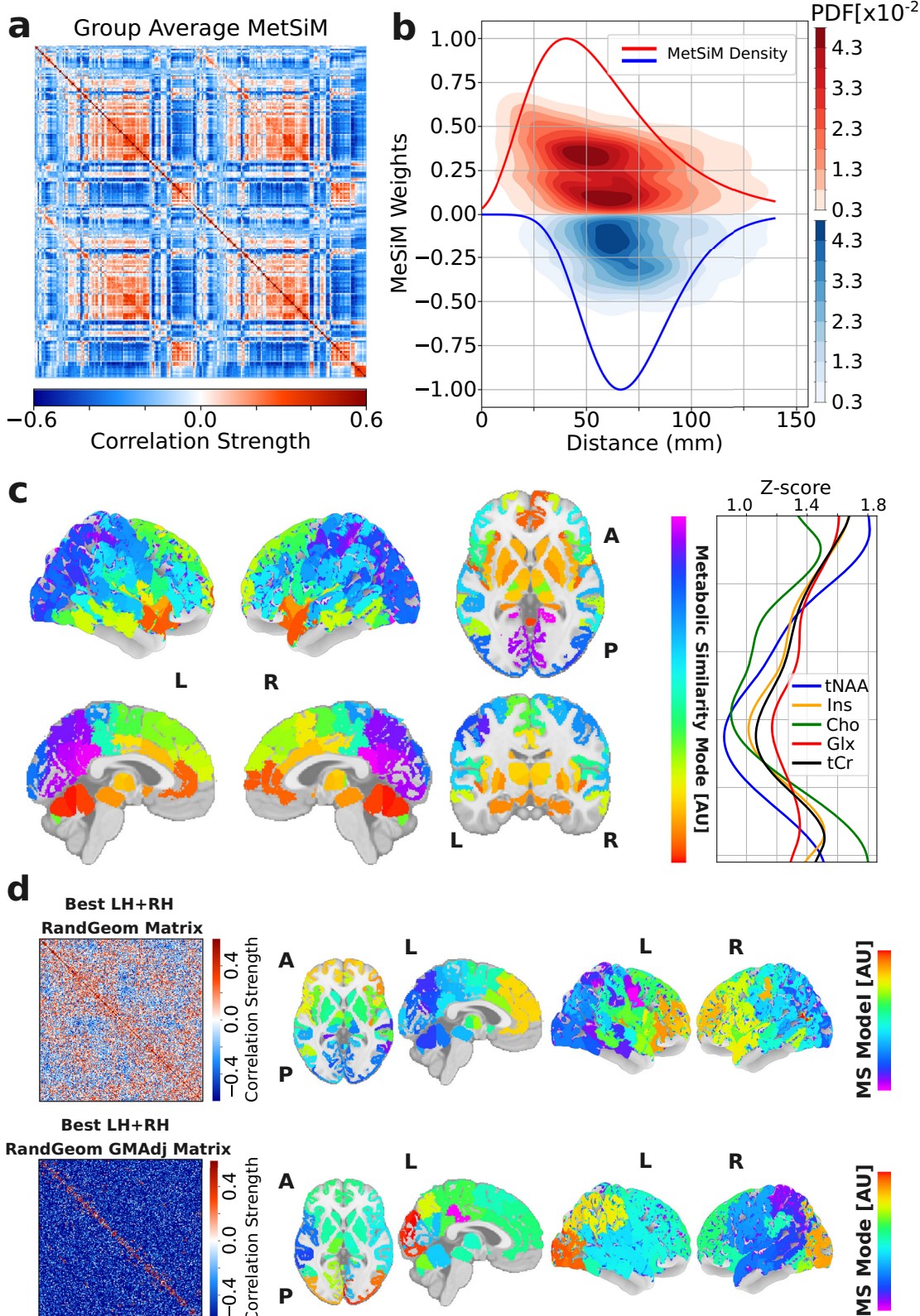

**Fig. 3 | Metabolic similarity mode. a** Group averaged (*N* = 51) MetSiM from the Geneva study. **b** Probability density functions (PDF) of negative (blue) and positive (red) metabolic pairwise correlations pooled from the group averaged (*N* = 51) MetSiM Geneva study aggregated across both hemispheres as a function of their respective inter-region Euclidean distance. Histogram values (z-axis) were smoothed using a kernel density estimation technique with Gaussian kernels. Continuous lines represent a lognormal fit over the estimated MetSiM density. **c** Spatial mapping of the group-averaged MetSiM matrix (Geneva study), reduced

to a single component using PCA-t-SNE, projected onto MNI space, and represented by the Metabolic Similarity Mode (MS mode). The MS mode is color-coded (color bar) and inverse-mapped to the five z-score-normalized metabolic profiles. **d** Synthetic MetSiMs (top: generated by a random geometric model; bottom: generated by a GM-constrained random geometric model GMAdj) yielding the highest correlation with the empirical MetSiM, along with their corresponding MS mode representations, derived as the first component of the PCA-t-SNE projection.

insula. The cerebellum is the only exception, metabolically aligning with the subcortical-prefrontal axis.

These systems, long recognized as functionally and structurally distinct, are here shown to be differentiated also in neurometabolic terms. This distinction is reflected in the spatial variation of the five metabolic profiles, which drive a monotonically increasing caudal-to-rostral MS mode and give rise to a recognizable gradient that we term the metabolic similarity gradient (MS gradient; schematized in Fig. 4c and formally defined in Methods Metabolic Principal Path Construction, Eq. (3)). Specifically, segregation is largely explained by a reversal in the tNAA-to-Cho ratio, tNAA dominating in the cortical regions and Cho in the subcortical regions, which underpins the observed inversion from predominantly positive MetSiM values (cortical-to-cortical correlations) to negative values (cortical-to-subcortical correlations).

## Topology, metabolic hubs and principal paths

We examined metabolic similarity network topology by thresholding individual MetSiMs (see Methods Weighted Matrix Binarization) to generate binary graphs and calculating rich-club coefficients across edge densities (2%, 10%, 18%, and 40%). At all densities, metabolic networks exhibited with rich-club coefficients being statistically significant ($p < 0.001$) when tested against both a degree-preserving null model and a spatially embedded random geometric model (see Methods Rich-Club and Supplementary Fig. 5). These findings were replicated on the group-averaged MetSiM (binarized MetSiM shown in Fig. 4a), and the resulting rich-club network mapped to the MNI coordinates is shown in Fig. 4b. Rich-club nodes were most concentrated in the occipital lobe and less so in the frontal lobe. Coloring nodes and edges by MS mode-derived values, revealed metabolically distinct and spatially segregated clusters–each containing at least one rich-club node, except the cerebellum.

To illustrate the integrative nature of these seemingly disparate high-similarity nodes, we leveraged the previously observed caudal-to-rostral MS gradient, which gives rise to a unique topological structure, and distilled it through the construction of a discrete analog of a principal curve[37,38]. Inspired by the original definition of principal curves as smooth, self-consistent trajectories through high-dimensional data, our approach identifies an optimal path within the metabolic network–constrained by gray-matter adjacency (GM constrained random geometric model aligns more closely with empirical MetSiMs), limited to individual hemispheres (which are metabolically homotopic), spatially aligning with local MS gradient progression and capturing the maximal variation of the MS mode distribution (see Methods Metabolic Principal Path Construction) between the lowest and highest MS mode nodes (see Fig. 4c). The resulting network paths were fitted with a piecewise cubic spline for visualization, 3D-rendered on Fig. 4d and schematically visualized in Fig. 4g.

The principal path, originating at the isthmus cingulate, first extends into the visual pericalcarine region, then traverses the lateral occipital cortex (LOC), continues through the lateral frontal lobe, and terminates in the rostral anterior cingulate cortex (rACC). A subcortical path, structurally disjoint yet metabolically connected via an edge between the rACC and caudate nucleus (interrupted path), links the thalamus, brainstem hubs, and the cerebellum. MS mode values vary smoothly and monotonically along both cortical and subcortical trajectories in each hemisphere (Fig. 4e). To test whether these paths reflect the joint influence of local spatial effects and global metabolic diversity, we compared them to 10,000 random MetSiM matrices (*RandGeom*), evaluating the normalized MS gradient ($\widetilde{\mathscr{G}}_{\mu}$, the sum of $L^2$ norms of local MS mode gradients along the path, normalized by the maximum attainable value for the same endpoints) and the normalized entropy ($\widetilde{H}$, path entropy relative to a maximal-entropy Erdős-Rényi graph) (see Methods Metabolic Principal Path Construction). The empirical paths jointly optimized $\widetilde{\mathscr{G}}_{\mu}$ and $\widetilde{H}$, with both occurring more extremely than expected under the spatial null model (joint one-tailed

permutation test: LH-CTX $\widetilde{\mathscr{G}}_{\mu}$ = 0.08, $\widetilde{H}$ = 0.79, $p$ = 0.0358; RH-CTX 0.09, 0.81, $p$ = 0.0110; LH-SUBC 0.11, 0.70, $p$ = 0.0030; RH-SUBC 0.13, 0.76, $p$ = 0.0005; Fig. 4f). This pattern indicates convergence towards high local metabolic similarity and high global metabolic diversity.

The schematic in Fig. 4h summarizes the spatial organization of metabolic similarity in the brain, as revealed by the principal path analysis. In our data, nearby adjacent gray matter regions tend to be metabolically similar, whereas distant regions are more metabolically diverse. This co-occurrence produces a coherent spatial structure that organizes into a traversing MS gradient, positioning the observed network topology between that of a highly structured lattice (low local variation, multiple gradients) and a random network (high global variability, no gradients). Importantly, these principal paths constitute measurable features of the metabolic network, allowing their properties to be quantitatively assessed and compared in subsequent analyses. The broader implications of this organization are considered in the Discussion.

## Metabolic similarity and structural connectivity

To test the hypothesis that MetSiM edges reflect axonal connectivity, we analyzed diffusion spectrum imaging (DSI) data from the Geneva study, reconstructing tractograms and deriving weighted structural connectivity matrices (SC) based on the same Chimera gray matter parcellation LFMIHIFS-3 (see Methods DSI and Structural Connectivity) to allow direct MetSiM and SC edge-wise comparisons (see Fig. 5a). After excluding individuals with poor quality tractograms, 42 SC matrices were retained, with only the nodes covered by both matrices used to produce 42 190 × 190 matrices.

A direct edge-wise Pearson correlation between structural connectivity and metabolic similarity yielded low scores (0.01 to 0.05, $p < 0.001$). We then deliberately extended the analysis to probe whether metabolic similarity between nodes could depend on indirect structural connections via intermediate nodes. By computing $\mathbf{S}^n$ (the $n$-th power of the SC matrix, see Methods Higher Order Connectivity), we found a peak correlation of 0.1 ($p < 0.001$) at second-order connections, which declined for higher orders (see Fig. 5b). Thus, despite the weak overall correspondence, metabolically similar nodes exhibit a very modest tendency to converge on the same structural targets.

We also tested whether perturbations of metabolic nodes could be influenced by distant structural nodes using communicability analysis[39]. Figure 5c shows correlations between the predicted communicability model and the observed metabolic network were consistently weak ($r < -0.022$, $p < 0.001$) in binary and weighted matrices. These findings indicate that the metabolic similarity network is effectively independent of structural connectivity under the tested communicability model.

Lastly, we applied the same principal path extraction procedure to the averaged structural connectivity matrix by first computing its first Uniform Manifold Approximation and Projection (UMAP) component, commonly referred to as the connectopy[40] (i.e., the spatial map of structural connectivity profiles; see Fig. 5d), and then deriving a path that aligns with its spatial gradient while maximizing the diversity of connectopic values along the trajectory. Despite the presence of smooth connectopic variation across brain lobes (LH-CTX $\widetilde{\mathscr{G}}_{\mu}$ = 0.16 and RH-CTX $\widetilde{\mathscr{G}}_{\mu}$ = 0.09), only disjoint paths could be identified, and no trajectory traversing all lobes was found–likely due to the limited diversity of structural connectivity profiles (LH-CTX $H$ = 0.63 and RH-CTX $H$ = 0.64). Cost function optimization results are shown in Supplementary Fig. 7.

Lastly, we tested the hypothesis that structurally central nodes exhibit higher metabolic activity. Spearman correlations between structural degree centrality and metabolic profiles (z-score normalized) (sample correlation shown in Fig. 6a) yielded values ranging from 0.19 to 0.25 across connection densities (1–20%) (see Fig. 6b). Taken together, these results confirm the poor resemblance between

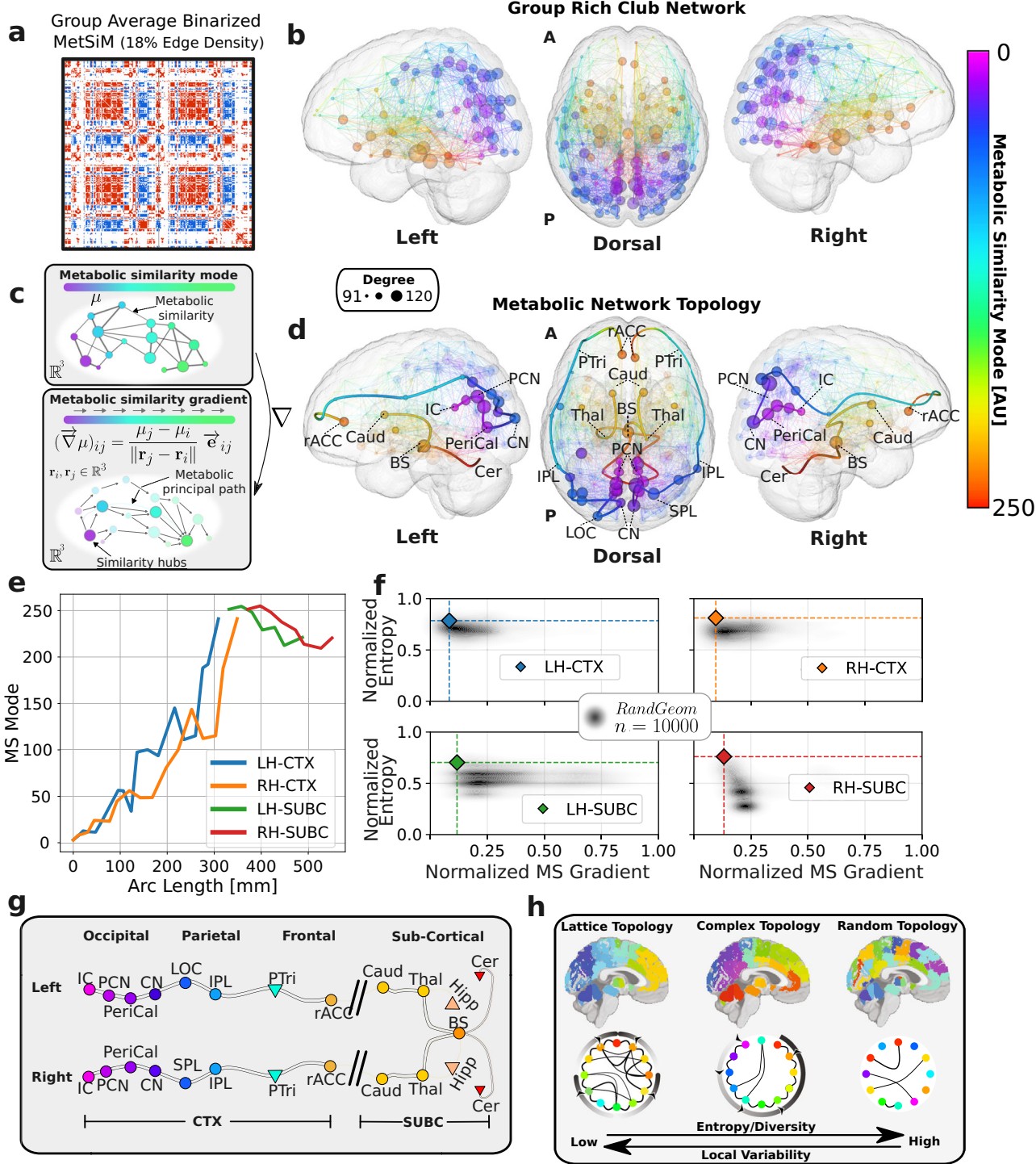

**Fig. 4 | Metabolic principal paths. a, b** Binarized Group MetSiM (18% edge density) from the Geneva study and associated rich club network. Rich-club nodes are shown at their anatomical locations, colored according to their MS mode values and with radii scaled by node degree. Interhemispheric edges are omitted for visualization purposes. **c** Schematic illustrating the MS mode (nodes colored by mode value) and the corresponding MS gradient (vector field, with arrow length inversely proportional to gradient norm, highlighting regions of minimal local variation). **d** Core topology of the metabolic connectome represented by similarity hubs (opaque) situated along the metabolic principal path depicting regions of metabolic similarity through a metabolic similarity gradient. Rich-club nodes and edges (transparent) are also displayed. **e** MS mode values for nodes along the two cortical (LH-CTX and RH-CTX) and two subcortical (LH-SUBC and RH-SUBC) principal paths. **f**, Joint null distribution of normalized MS gradient and Shannon entropy for observed paths compared to 1000 spatial null models. **g** Schematic

representation of the main metabolic principal path per hemisphere running through the cortex of four brain lobes and from subcortical structures to the brainstem and cerebellum. Similarity hubs (circles) and low similarity nodes (triangles) are highlighted along the paths. Metabolic similarity (but interrupted connection) between the subcortical fiber and the cortical fiber is indicated by a dashed line. **h** Scheme of three network topologies--lattice (left), complex (middle), and random (right)--illustrating the progression from multiple disjoint MS gradient tracks through a single traversing principal path to an unstructured gradient. Isthmus cingulate (IC), Precuneus (PCN), Pericalcarine cortex (PeriCal), Cuneus (CN), Superior parietal lobule (SPL), Inferior parietal lobule (IPL), Pars triangularis (PTri) of the inferior frontal gyrus, Rostral anterior cingulate cortex (rACC), Anterior insular cortex (aIns), Caudate nucleus (Caud), Thalamus (Thal), Brainstem (BS), Hippocampus (Hipp), and Cerebellum (Cer).

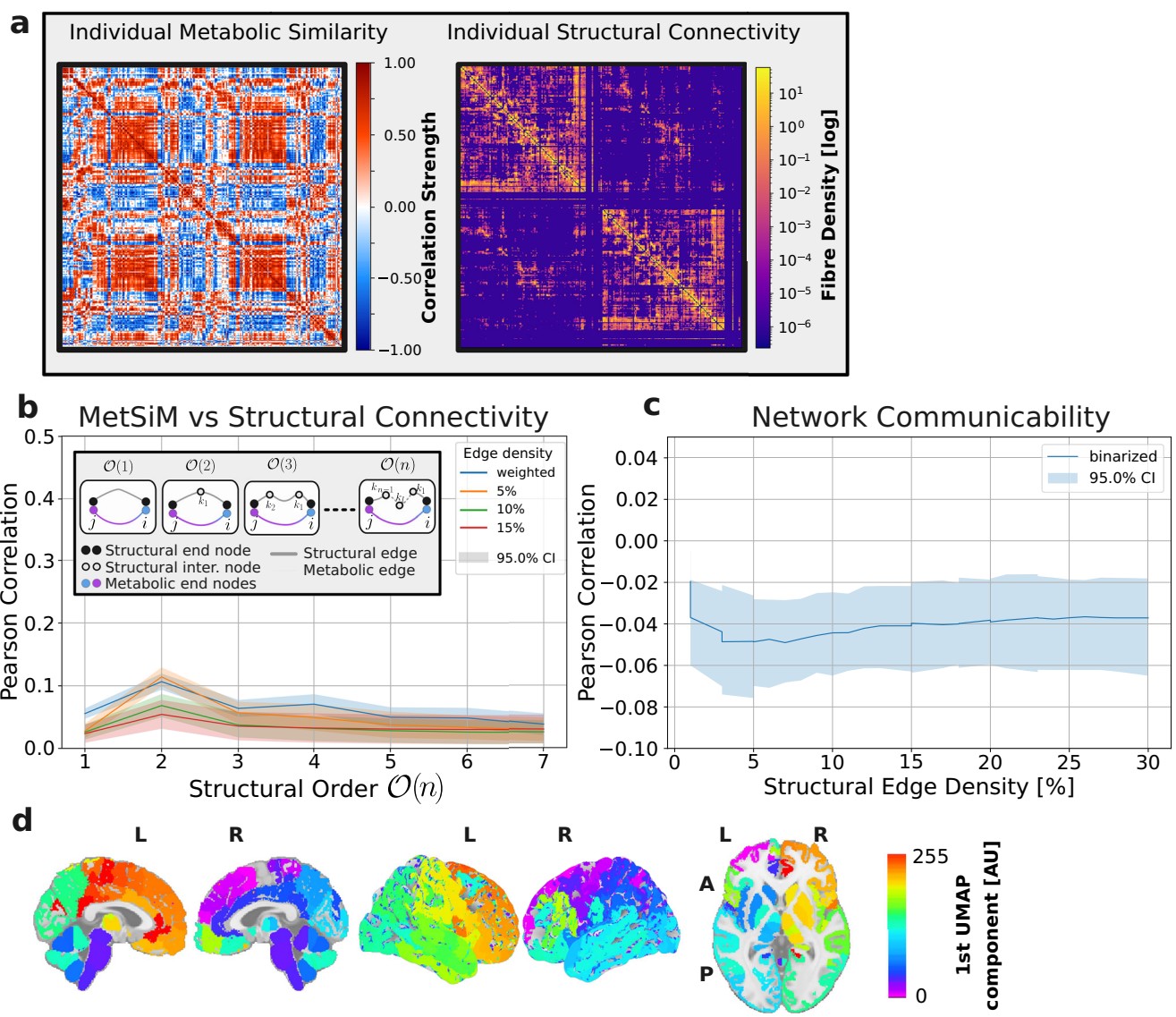

**Fig. 5 | Metabolic similarity and structural connectivity. a** Example individual metabolic similarity (MetSiM) and structural connectivity (SC; fiber density) matrices. **b** Edge-wise correlation between SC and MetSiM as a function of structural order $\mathcal{O}(n)$ (paths with $n - 1$ intermediary nodes). Points show the *mean* Pearson correlation across participants ($n = 42$); shaded bands indicate 95% confidence intervals. Two-sided tests used one-sample *t*-tests on Fisher-*z* transformed

correlations against zero. **c** Correlation between the communicability model $M = \exp(gS)$ and the observed MetSiM. Curves show the *mean* Pearson correlation across participants with 95% confidence intervals; two-sided one-sample *t*-tests on Fisher-*z* values assessed deviation from zero. **d** First uniform manifold approximation and projection (UMAP) component of group average connectivity matrix from the Geneva study.

structural connectivity and metabolic similarity but establish a link between structural centrality and high $^1$H-MRSI metabolite levels.

### Metabolic similarity and cytoarchitecture

To test the hypothesis that metabolic networks align with cytoarchitectonic similarity networks, we analyzed three datasets: the Cognitive-Consilience dataset, the Von Economo dataset, and the BigBrain dataset (see Methods Cytoarchitecture).

The Cognitive-Consilience dataset, based on cortical laminar patterns[41], was digitized by Seidlitz et al.[35] and extended in this study to include subcortical and cerebellar regions (see Methods Cognitive Consilience Classification), comprising nine bilaterally symmetric cytoarchitectonic classes (see Fig. 7a). We applied Gaussian Mixture clustering with nine clusters to the Geneva study group's averaged MetSiM MS mode map and compared the resulting clusters to the cytoarchitectonic classes. Alignment between the clusters and cytoarchitectonic classes was assessed using overlap metrics and

spatial permutation tests (see Methods Inter-Class Overlap Score). Individual MetSiMs yielded a significant overlap score of 0.30 ± 0.02 (AdjPerm $p = 0.008$), and the group average MetSiM showed a higher score of 0.36 (AdjPerm $p < 0.001$). Cluster correspondence analysis revealed strong alignment in the primary somatosensory cortex (class 1 and metabolic class 7, overlap: 0.632), subcortical regions (class 8 and metabolic class 1, overlap: 0.577) and in the premotor cortex (class 2 and metabolic class 6, overlap: 0.730). Full results are reported in Supplementary Table 4.

The Von Economo dataset included 42 bilaterally symmetric cortical cytoarchitectonic profiles mapped onto the 210-node LFMI-HIFS-3 parcellation (left Fig. 7b). A direct edge-wise comparison between the group-average MetSiM and the Von Economo similarity matrix (see Methods Cognitive Consilience Classification) yielded a Spearman correlation of 0.09 (AdjPerm $p < 0.001$) which dropped to 0.05 ($p < 0.001$) after controlling for spatial autocorrelation. This global association was primarily driven by the occipital lobe, medial

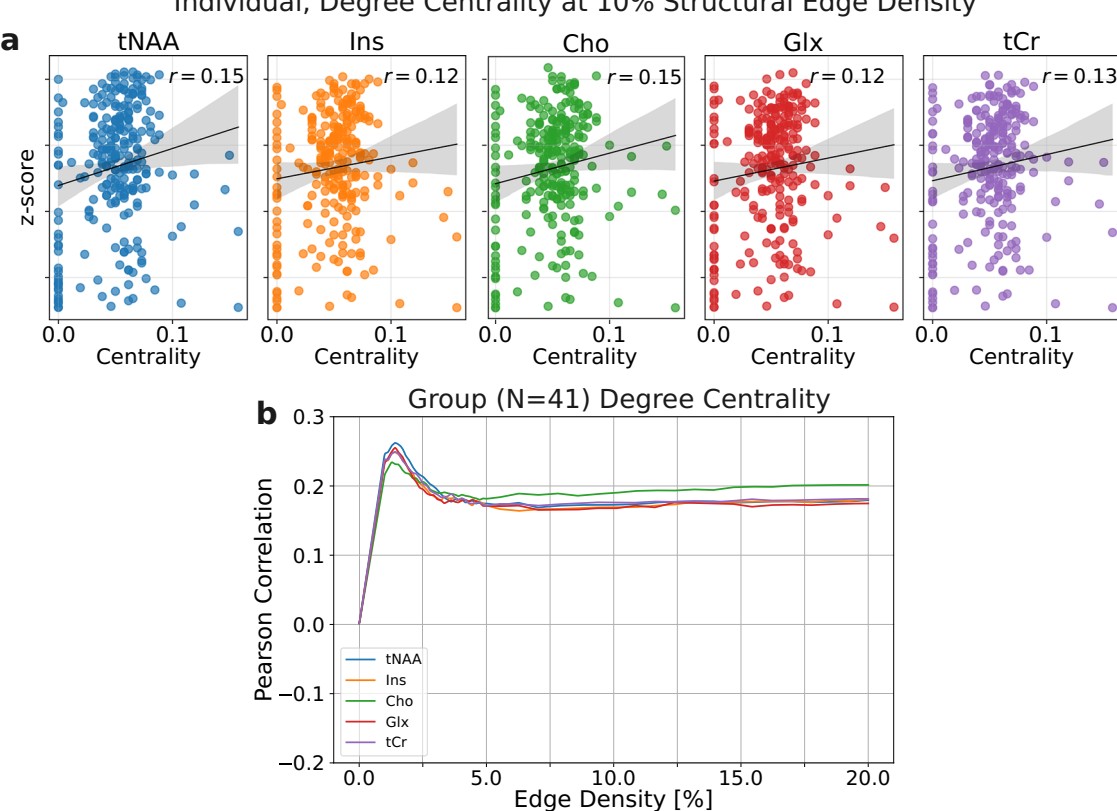

**Fig. 6 | Metabolic levels and structural connectivity. a** Example participant: scatter of z-scored ¹H-MRSI metabolite levels versus structural node degree (10% edge density). Line shows the ordinary least-squares fit; the shaded band is the 95% confidence interval of the fitted line (bootstrap CI). Spearman's $\rho$ ($p < 0.001$) are reported for association. **b** Group results ($n = 41$): Spearman correlation between z-scored metabolite levels and degree centrality across structural edge densities. Points show the mean correlation across participants; error bars indicate 95% confidence intervals.

superior prefrontal regions, precentral gyrus, superior temporal gyrus, and posterior insula ($r = 0.45 - 0.51$) as shown in the absolute nodal correlation metric on Fig. 7b (see Methods Absolute Nodal Correlation). In contrast, the anterior cingulate cortices showed minimal correlation.

The BigBrain similarity matrix (left Fig. 7c)[42], derived from ultra-high-resolution cytoarchitectural data of the cortex[43] (see Methods Big Brain Similarity Matrix), shows a node-wise correlation of 0.29 (AdjPerm $p < 0.001$) with the group-average MetSiM and dropped to 0.26 ($p < 0.001$) after correcting for spatial autocorrelation. The highest absolute nodal similarities were observed in the occipital lobe, postcentral gyrus, superior frontal gyrus (medial part), caudal-medial superior frontal cortex, and posterior cingulate cortex ($r = 0.45-0.51$). Similar to the Von Economo similarity matrix, the anterior cingulate cortex showed minimal correlation.

Lastly, we applied the principal path extraction procedure to the Von Economo and BigBrain similarity matrices by computing their first PCA-t-SNE component to obtain a cytoarchitectonic similarity index (CSI) (bottom panels of Fig. 6b,c) and attempting to derive a path minimizing the normalized MS gradient and maximizing normalized entropy. No valid path was identified, likely due to a weak continuous similarity gradient (LH/RH-CTX Von Economo $\widetilde{\mathscr{G}}_{\mu} = 0.12$; BigBrain 0.07) visible in Fig. 7b,c, in the presence of a high CSI diversity (LH/RH-CTX Von Economo $\widetilde{H} = 0.84$; BigBrain LH-CTX 0.86, RH-CTX 0.76). Cost function optimization results are provided in Supplementary Fig. 7.

These results suggest that MetSiM topology partly reflects the spatial organization of cytoarchitectonic patterns, and is primarily driven by the high diversity of distinct metabolic similarity profiles across the cortex.

## Metabolic similarity and genetic co-expression

We next tested our second biological hypothesis, that MetSiM edges are characterized by high levels of gene co-expression, by leveraging the nearly complete human genome dataset (20,737 genes) from the Allen Brain Institute. Using data from six adult human post-mortem brains[44], we mapped whole-genome transcriptional profiles onto the same LFMIHIFS-3 parcellation used to define the MetSiM's 277 nodes (see Methods Genetic Co-Expression). This approach allowed us to estimate inter-regional co-expression patterns (see Fig. 8) for every possible pair of nodes within the same anatomical reference frame as the MetSiM. A significant positive correlation was observed between MetSiM edge weights and whole-genome co-expression values (PermAdj $r = 0.34$, $p < 0.001$) and $r = 0.32$ ($p < 0.001$) when accounting for spatial autocorrelation, indicating that metabolically similar regions also share similar gene expression profiles. Regions in the occipital lobe and superior frontal cortex (medial) showed the strongest absolute nodal correlation ($r = 0.60 - 0.74$), while the anterior cingulate cortex, brainstem and cerebellar regions contributed the least. This strong alignment between metabolic and genetic similarity is reflected in the principal path of genetic similarity from a PCA-t-SNE reduction (Fig. 8c and optimization results show in Supplementary Fig. 7), which follows a similar caudal-to-rostral trajectory as the MetSiM path (Fig. 8e; LH/RH-CTX $\widetilde{\mathscr{G}}_{\mu} = 0.07$; LH-CTX $\widetilde{H} = 0.88$, RH-CTX 0.83). This concordance, and the topological prerequisites for such paths to emerge, suggest that the local and global effects characterizing metabolic organization may, in part, be shaped by underlying transcriptional architecture. Furthermore, after a gene ontology (GO) analysis of the 20,737 genes, we extracted 250 GO categories with brain-enriched contributions (FDR-corrected enrichment ratio > 1). A

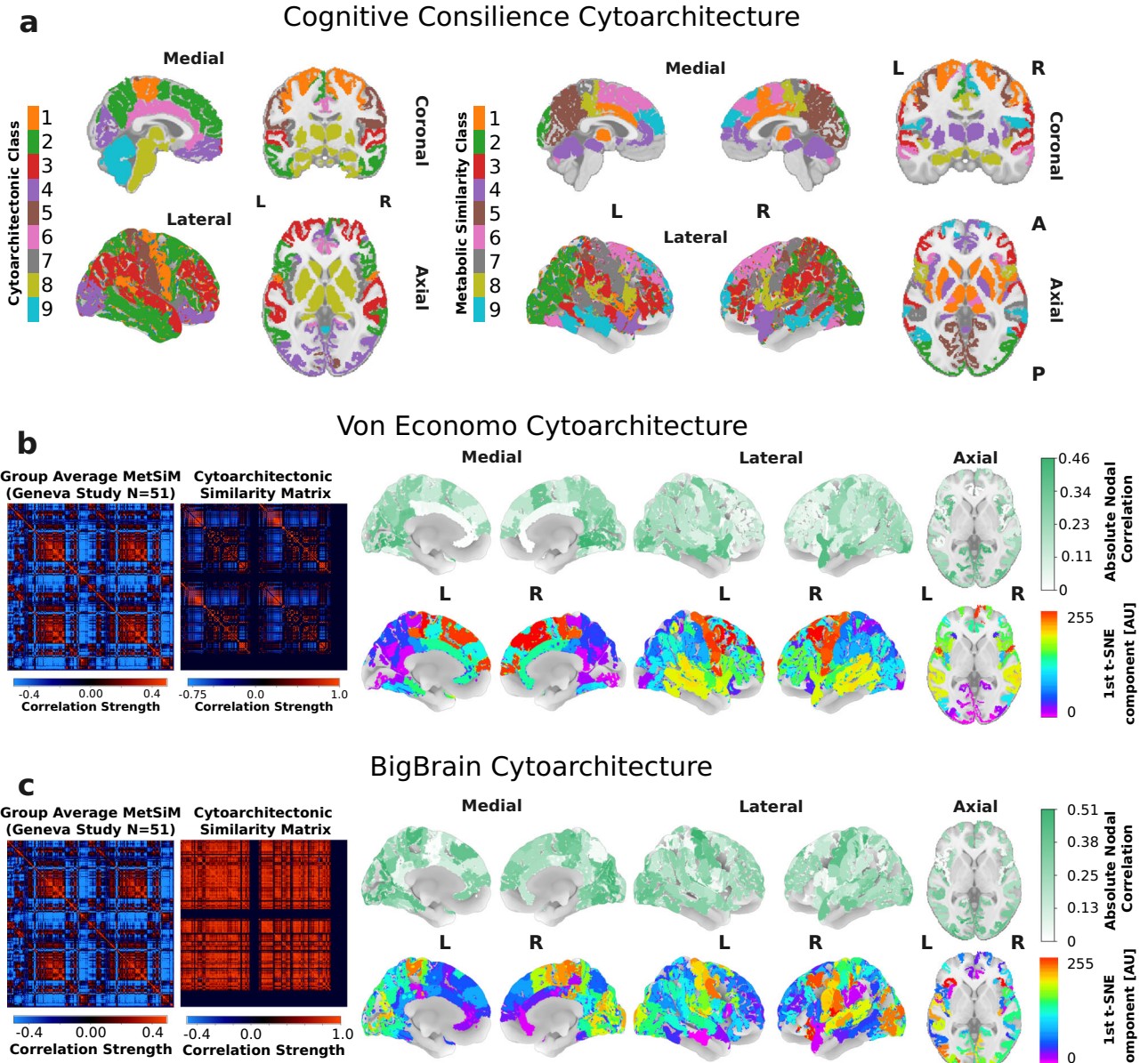

**Fig. 7 | Metabolic Similarity and Cytoarchitectural Similarity.**
**a** Cytoarchitectonic classification of cortical laminar patterns performed by Seidlitz et al.[35] (left) and based on the Cognitive-Consilience dataset[41] and classification of metabolically similar cortical regions based. **b** Von Economo cytoarchitectural similarity matrix, its corresponding absolute nodal correlation with the group MetSiM ($p < 0.05$; AdjPerm) (top) and its first PCA/t-SNE component (bottom).

**c** BigBrain cytoarchitectural similarity matrix and its corresponding absolute nodal correlation with the group MetSiM ($p < 0.05$; AdjPerm) (top) and its first PCA/t-SNE component (bottom). *Statistics (b,c)*. Global and nodal associations were tested with two-sided Spearman rank correlations. Significance was derived from 10000 permutations; adjusted permutation $p$-values are reported as AdjPerm.

leave-one-GO-class-out analysis, identified, via the elbow method, 17 parent GOs that contributed the most (see Supplementary Fig. 8). Most of these contributing categories are related to developmental processes (e.g., cell division and polarity, cell migration, neuron projection development), transport (e.g., inorganic cation and anion, amino acids, organophosphate ester), and cell-cell signaling including synaptic signaling and transmission. Multicellular organismal-level homeostasis and ensheathment of neurons by glia cells are also among the most contributing GO categories. By contrast, the least contributing GO categories include embryo development, development of organs not directly related to the brain, organelle assembly, RNA metabolic and catabolic processes. Full results are reported in Supplementary Data 2 and 3. These results indicate that MetSiM topology aligns with spatial expression patterns of brain-expressed genes that

play a central role in supporting brain development and function as well as homeostasis.

## Discussion

We have demonstrated how MetSiMs, derived from 3D high resolution whole-brain ¹H-MRSI, can be used to estimate the metabolic similarity between brain regions in a sample of healthy participants at both the individual and group levels. We developed a reliable and consistent method for deriving MetSiMs, which we have shown to be also replicable in an independent sample scanned using a different MRI scanner. Furthermore, we recurred to a brain metabolic network analysis, which enabled us to first investigate the biological basis of the brain metabolic organization and test the resemblance of the MetSiMs-derived network with other known brain networks.

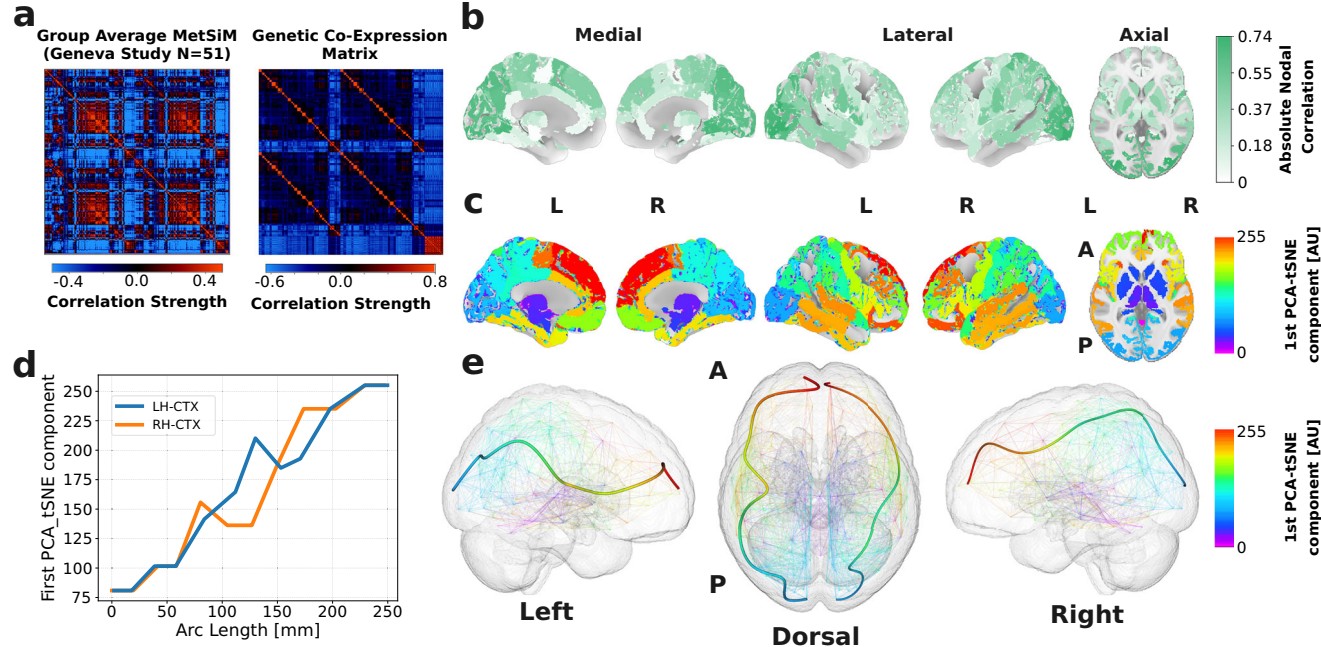

**Fig. 8 | Metabolic Similarity and Genetic Co-expression. a** Genetic co-expression matrix derived from the Allen Human Brain Atlas; **b** corresponding absolute nodal correlation with the group MetSiM ($p < 0.05$; AdjPerm); **c** first PCA/t-SNE component of the co-expression matrix. **d** Nodal PCA/t-SNE values along the two cortical principal paths (LH-CTX and RH-CTX); **e** spatial mapping of their centroids on an MNI brain. Principal Component Analysis (PCA), t-distributed Stochastic Neighbor Embedding (t-SNE). Global and nodal associations were tested with two-sided Spearman rank correlations. Significance was derived from 10000 permutations; adjusted permutation $p$-values are reported as AdjPerm.

We observed that MetSiM distribution follows an inverse relationship with distance: spatially proximate nodes are metabolically similar, whereas distant nodes are metabolically dissimilar. This distance-dependent organization parallels patterns reported in other brain networks, where short-range anatomical connections tend to be stronger and more prevalent, while long-range connections are sparse and weak[45]; similarly, positive functional correlations typically occur at short distances, whereas anti-correlations are more common between distant regions and antagonistic networks[46]. Such spatial dependencies are characteristic of the complex topological organization observed in naturally embedded networks[47], where modular architectures balance local clustering with global integration. Beyond this spatial tendency, MetSiMs also align along functional dimensions, showing clustering of functionally similar regions (e.g., the limbic system) and segregation of functionally distinct regions (e.g., primary somatosensory vs. primary motor cortices). Notably, this pattern is primarily driven by a decrease in the tNAA/Cho ratio (relative to colinear levels of tCr, Glx, and Ins) from the occipital lobe (high ratio) through the parietal and frontal cortices, extending into subcortical structures, the brainstem, and the cerebellum (low ratio).

This decrease is consistent with existing MRS literature, which demonstrates higher tNAA (a neuronal density marker) in gray matter-rich cortical regions, and lower tNAA with relatively higher Cho (a membrane turnover marker) in subcortical structures such as the basal ganglia and brainstem[48–50]. Even after partial volume correction, regional variation persists: cortical GM–particularly in the occipital lobe–has a higher density of mature neurons[51] and lower glial content than subcortical GM[52], which is richer in glial cells, including a higher density of oligodendrocytes[53], and exhibits greater myelin-related membrane dynamics. The observed decrease in tNAA/Cho ratio from the occipital lobe to subcortical structures thus reflects a shift from neuron-rich to glia-enriched regions with elevated membrane turnover. Of note, one GO categorie that contributes to the correlation between MetSiM and gene co-expression is related to ensheathment of neurons by glia cells.

Topographic organization of network properties is common in biology, with one of the most striking examples being the tonotopy along the basilar membrane of the cochlea. There, a monotonic graded change in membrane properties (stiffness, width, and mass distribution) maps low- to high-frequency responses of outer hair cells along the cochlear spiral[54]. Analogously, the MS gradient revealed by the metabolic principal path shows a spatially ordered, monotonically increasing progression of MS mode values along the caudo-rostral axis, mapping distinct functional brain specializations into a continuous biochemical space. Albeit the occurennce of such a gradient in other biological system, we do not claim that the resulting metabolic principal paths have a direct physical or biological instantiation; rather, they represent measurable signatures of a network topology balancing local clustering and global diversification. Similar to structural brain networks, where nearby gray matter regions form dense connections to minimize wiring cost and sparse long-range links preserve global efficiency[6], metabolic connectomics appears to balance local metabolic variability with global metabolic diversity. Minimizing the former creates a coherent spatial structure on an otherwise random topology, while maximizing the latter organizes this structure into a traversing MS gradient with a monotonic MS mode. This *economy of metabolic distribution* is reminiscent of the small-world topology in structural networks[6], where such balance enables efficient communication routes and, beyond its biophysical interpretation, has served as a marker of network integrity in health and disease[7,8]. By analogy, we propose that the occurrence of metabolic principal paths reflects a direct measure of metabolic coherence in brain networks.

Large-scale spatial gradients as continuous patterns of variation in brain organization have been found across structural, functional, microstructural, transcriptomic, and neurotransmitter receptor data[55]. Constant, whole-brain-traversing rostro-caudal gradients (resulting from monotonically increasing modes) are rarely reported in the literature, likely because most brain organization follows more complex, multidirectional patterns and is typically confined to specific brain systems, such as the principal morphometric similarity mode

separating motor from sensory cortices[56], the developmental and evolutionary cortical expansion where association cortices enlarge most relative to primary areas[57], or the microstructural "sensory-fugal" gradient that extends from sensory anchors toward transmodal cortex[58]. The study most closely aligned with our results is that of Oldham et al.[59], who combined in vivo MRI-based structural connectivity with post-mortem gene expression data to identify a medial-lateral thalamic gradient mapping onto an anterior-posterior (rostro-caudal) cortical pattern. We observed a congruent anterior-posterior gradient in our MS gradient, obtained purely from non-invasive in vivo ¹H-MRSI data. In this context, the thalamus–identified by Oldham et al. as the driver of the thalamo-cortical gradient and recognized as a central structural and functional hub, may likewise act as the orchestrator of metabolite diversity, assembling locally coherent metabolic similarity patterns into the complex topology observed at the macroscale. We found that MetSiMs show only weak correspondence with structural connectivity–both directly, where the presence of a structural edge does not imply metabolic similarity, and indirectly, where shared neighbors mediate similarity only to a limited extent. Instead, their organizational structure is better captured by gradients: locally, MS modes align with connectopies, reflecting high proximal similarity, but globally they diverge, as metabolic similarity rarely extends over long distances. By contrast, connectopies display redundancy both proximally (neighboring nodes sharing similar profiles) and distally (distant nodes converging on the same set of hubs). As hypothesized, cortical areas similar in MetSiM networks also exhibit moderate cytoarchitectural similarity and high genetic co-expression, consistent with the established link between shared cellular machinery and parallel metabolic demand[33,42,60,61]. Notably, the genetic co-expression gradient not only follows the same rostro-caudal trajectory as the MS gradient but also suggests that both the local and global properties of MS are shaped by underlying genetic organization. Taken together, these results indicate that metabolic similarity networks provide a biologically grounded, node-centric organizational framework that is locally consistent with but globally distinct from the white matter edge-centric connectome.

Building on these previous observations, we propose that the metabolic principal path extends beyond a usefull mathematical construction, and it is a remnant of the neurodevelopmental processes governing the maturation of the neural tube and its subsequent unfolding into subcortical and cortical structures. During early neurodevelopment, the neural tube differentiates into three primary divisions from rostral to caudal; the prosencephalon (forebrain), mesencephalon (midbrain), and rhombencephalon (hindbrain), preserving a sequential arrangement such that initially adjacent regions remain proximal in the mature central nervous system. While each region adopts a distinct gene expression profile that guides its development, function, and metabolism, the extent of its spatial variation remains constrained by proximity effect: adjacent areas share similar microenvironments[62], exhibit local diffusion of signaling molecules (morphogens)[63], and display aligned epigenetic modifications (e.g., DNA methylation patterns)[64]. Spatially assimilated regions share similar gene expression, hence produce comparable metabolic enzymes and display analogous metabolic activity[61]. Consequently, proximity in the neural tube translates into adjacency in the mature metabolic principal path, yielding its unbroken continuity observed from the forebrain through the midbrain to the hindbrain. Consistent with this interpretation, the GO categories that most strongly contribute to the MetSiM-gene co-expression association are enriched for neurodevelopmental processes.

Our findings show that variation in ¹H-MRSI-measured metabolite levels across high-centrality nodes predicts the structural backbone of the human connectome, suggesting that metabolic organization may precede and support structural connectivity. Whereas PET–particularly FDG-PET–has demonstrated associations between glucose metabolism and structural centrality[13,65,66], ¹H-MRSI captures a broader spectrum of cellular processes, including neuronal integrity (tNAA), energy buffering and storage (tCr), excitatory neurotransmission (Glx), membrane turnover (Cho), and glial activity (Ins)[67], providing a richer biochemical basis for network inference. Moreover, compared to PET, the proposed ¹H-MRSI-based metabolic connectome offers key advantages: it is non-invasive, avoids ionizing radiation, can simultaneously quantify multiple metabolites in a single acquisition (8-20 minutes depending on protocol, field strength and acceleration[26]), and provides direct insight into neurochemical pathways beyond energy metabolism. In contrast, PET and hybrid PET/MRI approaches, while powerful for mapping glucose or receptor binding[68–72], remain limited by long scan times, higher motion sensitivity, cost, neurovascular coupling effects, and the use of radioactive tracers[73,74].

This study has several limitations to consider. MRSI's limited coverage excludes orbitofrontal and basotemporal regions due to susceptibility artifacts and signal distortion. Additionally, the 5 mm isotropic spatial resolution challenges the accurate assessment of smaller subcortical regions. As a result, the MetSiMs constructed here may lack critical information, potentially omitting key network structures like the default mode network located in these areas. Furthermore, the constructed MS mode map in this study was based on group-averaged data, which overlooks inter-individual variability. Factors such as participants' resting or task states could influence metabolic similarity and were not addressed here. Future work with larger multi-institutional cohorts will allow a more systematic assessment of generalizability and the influence of neurological and psychiatric conditions on MetSiMs. Although we have not yet evaluated the alignment between MetSiMs and functional connectivity, our findings reveal that these networks possess a functional modular structure, which suggests that a strong correlation may exist.

In summary, our work provides a topological analysis of whole-brain gray matter metabolic similarity networks, made possible by recent advances in ¹H-MRSI. Beyond replicating canonical principles of brain network organization, we show that the observed topological properties can be directly linked to regional metabolite variation and their interplay, shaped by genetic and cytoarchitectural constraints, thereby establishing metabolic similarity as a biologically meaningful proxy for underlying cellular and molecular processes. Through their association with neurodevelopmental processes, such networks may play a fundamental role in shaping both structural and functional brain systems. This highlights their potential as biomarkers, capable of revealing metabolic disruptions linked to disease even before structural or functional changes occur or symptoms manifest. Comparing these networks between controls and individuals at risk for disorders such as schizophrenia could provide deep insights into underlying pathological mechanisms and identify metabolic alterations affecting central brain regions prior to the onset of the first psychotic episode. Thus, the development of MRSI-based metabolic networks should be viewed not merely as a technical advance but as a practical tool poised for integration into a comprehensive suite of clinical applications.

## Methods

This research complies with all relevant ethical regulations. This study is a secondary analysis of de-identified MRI/MRSI data from two independent cohorts that were conducted previously and approved by their respective ethics committees: the Geneva Cantonal Research Ethics Committee (CCER; protocol 2018-01731) and the Cantonal Research Ethics Commission on Human Research, Canton of Vaud, Switzerland (CER-VD; protocol PB 2017-00675 82/14). Data handling complied with applicable data-protection laws and institutional policies; only de-identified data were used and no re-identification was attempted.

## Groups

**Geneva Mindfulteen study.** Participants were recruited and scanned in the context of the Mindfulteen study, a randomized controlled trial to assess the effects of a mindfulness-based intervention on adolescent well-being. Details on recruitment, inclusion, and exclusion criteria can be found in the published protocol[34]. Briefly, the study recruited adolescents ($n = 69$, sex assigned at birth: 39 female, 30 male, gender not assessed, age range = 13–15 years) from the general population. Exclusion criteria included chronic somatic diseases or significant medical conditions, recent psychotherapy (< 6 months), recent use of psychotropic medications (<6 months), and any history of psychiatric disorders, except for current anxiety disorders or a past episode of major depressive disorder resolved at least 6 months earlier. The study protocol was approved by the Geneva Regional Ethical Committee (CCER 2018-01731). All participants (assent) and their legal guardians (consent) provided written informed consent/assent in accordance with the Geneva Regional Ethics Committee. No a priori sex- or gender-based analyses were planned given the study's scope and sample size.

**Lausanne psychosis cohort.** For the replication of our results in an independent sample scanned at a different site, we included 13 healthy controls (sex assigned at birth: 4 female, 9 male; gender not assessed, age range = 15-35 years), from the Lausanne Psychosis Cohort[36]. They were assessed by the Diagnostic Interview for Genetic Studies[75] to exclude a major mood, psychotic, or substance use disorder or had a first-degree relative with a psychotic disorder. Written informed consent was obtained from adult participants; for minors, written parental/guardian consent and participant assent were obtained. The study protocol was approved by the Cantonal Ethics Committee for Research on Human Beings (PB 2017-00675 82/14).

## MRI image acquisition

For the Geneva study, magnetic resonance (MR) data were acquired using a 3-Tesla scanner (Magnetom TrioTim, Siemens Healthineers, Forchheim, Germany) equipped with a 32-channel head coil at the Brain and Behavior Laboratory in Geneva. Each scanning session included a magnetization-prepared rapid acquisition gradient echo (MPRAGE) T1-weighted sequence with an in-plane resolution of 1 mm and a slice thickness of 1.2 mm, covering a volume of $240 \times 257 \times 160$ voxels. The repetition time (TR), echo time (TE), and inversion time (TI) were set to 2300 ms, 2.98 ms, and 900 ms, respectively. Additionally, a diffusion spectrum imaging (DSI) sequence was performed, acquiring 128 diffusion-weighted images with a maximum b-value of 8000 s/mm² and one $b_0$ reference image. The acquisition volume for DSI consisted of $96 \times 96 \times 34$ voxels with a resolution of $2.2 \times 2.2 \times 2.2$ mm. The TR and TE for the DSI sequence were 6800 ms and 144 ms, respectively.

The 3D $^1$H-FID-MRSI sequence accelerated by compressed-sensing[24] was acquired with 1.50 ms TE, 372 ms TR and 35 deg flip angle. The Field-of-View (FoV) size was $210 \times 160 \times 105$ mm (anterior-posterior, right-left, head-foot directions) with a 95 mm-thick slab selection, with a spatial resolution of $5 \times 5 \times 5.3$ mm and the spectral bandwidth was 2 kHz acquired with 512 points. The reference water acquisition had the following parameters: same TE, TR of 25 ms, lower flip angle of 3°, same FOV size, lower resolution of 6.6 x 6.7 x 6.6 mm, same bandwidth and free induction decay (FID) size of 16 points.

For the Lausanne Psychosis Cohort, MRI sessions were conducted on a 3-Tesla scanner (MAGNETOM Prisma fit, Siemens Healthineers, Forchheim, Germany) at the Lausanne University Hospital for 13 healthy controls. While the MPRAGE and DSI sequences parameters were identical to those used in the Mindfulteen study, the the 3D $^1$H-FID-MRSI sequence had different parameters than the one implemented for the Geneva study (TE = 1.0 ms, TR = 353 ms and flip angle of 40 deg). The FoV was the same as in Lausanne, with the same spatial resolution of $5 \times 5 \times 5.3$ mm, as the spectral bandwidth and FID size.

The water acquisition had the following parameters: same TE of 1.07 ms, TR of 25 ms, and a flip angle of 5 deg. FOV size, resolution, bandwidth and vector size were the same as for the Geneva study.

Complete details on MRSI acquisition, reconstruction, quantification, and validation are given in the original methodological reference[24]; MRSI parameters are summarized in Supplementary Table 5 following the MRSinMRS checklist[76].

## Preprocessing

**Anatomical.** For each subject, the anatomical T1-weighted image was used to parcellate the brain into different regions using the FreeSurfer package (version 7.2.0, http://surfer.nmr.mgh.harvard.edu) and subsequently identify nine distinct supra-regions: cortex, basal ganglia, thalamus, amygdala, hippocampus, hypothalamus, cerebellum, brainstem, and white matter.

We relied on the Chimera parcellation software tool[77] to create a combined volumetric parcellation for each input subject anatomical image based on the following atlases (as listed in the original Chimera repository) and referred to as the LFMIHIFS-3 parcellation scheme:

- **L:** Lausanne cortical parcellation scale 3[78],
- **F:** Basal ganglia parcellation[79],
- **M:** Thalamus parcellation[80],
- **I:** Amygdala parcellation[81]
- **H:** Hippocampus parcellation[82]
- **I:** Hypothalamus parcellation[83],
- **F:** Brainstem parcellation[82],
- **S:** Cerebellum parcellation[84],

**DSI and structural connectivity.** The DSI volumes were visually inspected for signal loss, which could indicate motion artifacts and potential exclusion from the analysis[85]. The DSI data were reconstructed following the methodology described by Wedeen et al.[86]. Preprocessing, anatomically constrained tractogram construction using deterministic tractography, and structural connectivity computation were performed using the MRtrix3 software tool[87].

Particularly, the peaks of the local fiber orientation distribution functions (fODFs) (maximum of three peaks per fODF) were computed and served as input for the deterministic tractography[88]. The tractography parameters included a maximum track length of 300 mm, 32 seeds per white matter voxel per diffusion direction, and termination criteria of a 60-degree angle between subsequent propagation steps or reaching the white-gray matter boundary. To ensure reproducibility, $10^6$ streamlines were selected[89].

A weighted, undirected brain connectivity matrix was reconstructed for each subject by filtering the whole-brain tractogram using Spherical-deconvolution Informed Filtering of Tractograms (SIFT)[90], which refined streamline contributions based on their alignment with underlying diffusion data and termination coordinates. We defined the sub-regions of the Chimera parcellation (see Section Anatomical) as the nodes of the structural connectivity. The strength of the connections $S_{ij}$ between any two node $i$ and $j$ were computed as the number of streamlines $ns_{ij}$ connecting these two regions weighted by the inverse of their average length $\frac{1}{l_{ij}}$ and the inverse of the node's volumes $\frac{1}{V_i + V_j}$ i.e., $S_{ij} = \frac{1}{l_{ij}} \cdot \frac{1}{V_i + V_j} \cdot ns_{ij}$. The length and volume terms were used to eliminate the bias towards longer fibers introduced by the tractography algorithm and the bias for the variable size of cortical ROIs, respectively[45]. For consistency and to limit possible false-positive connections, the connections that present in less than 50% of participants were discarded[91,92].

**MRSI reconstruction.** The MRSI data was reconstructed using a low-rank model constrained by total-generalized variation, with prior removal of subcutaneous lipid contamination and residual water signals[25]. Following reconstruction, the spatio-spectral data was analyzed using LCModel[93] to quantify metabolite signal in each voxel,

using the water signal from the additional acquisition as a reference. Because of the ultra-short echo time (1.5 ms), no $T_2$ relaxation corrections were required[25]. Although $T_1$ weighting is present because of the reduced TR and would differentially affect GM and WM, only GM-GM correlations were considered in this study. Thus, explicit $T_1$ correction would merely introduce a global scaling factor without altering the reported results. However, the 5 mm isotropic resolution may introduce partial signal leakage across GM-WM boundaries; therefore, potential GM-WM composition effects were further mitigated using voxel-wise partial volume correction in combination with a MRI point-spread-function correction (see Methods Section MRSI Partial Volume Correction).

The basis set for LCModel fitting included the following metabolites: N-acetylaspartate (NAA), N-acetylaspartylglutamate (NAAG), creatine (Cr), phosphocreatine (PCr), glycerophosphocholine (GPC), phosphocholine (PCh), myo-inositol (mI), scyllo-inositol (sI), glutamate (Glu), glutamine (Gln), lactate (Lac), gamma-aminobutyric acid (GABA), glutathione (GSH), taurine (Tau), aspartate (Asp), and alanine (Ala). Only a subset of these metabolites could be reliably resolved and due to overlapping spectral peaks, certain metabolites were combined: NAA and NAAG (denoted tNAA), Cr and PCr (denoted tCr), GPC and PCh (denoted Cho), mI (denoted Ins), and Glu and Gln (denoted Glx), resulting in five distinct metabolite volumes. LCModel also provided spectral quality metrics, such as the signal-to-noise ratio (SNR), Cramer-Rao Lower Bound (CRLB) for each metabolite estimation.

**MRSI registration.** The MRSI tCr volume was co-registered to the corresponding T1-weighted anatomical image using Advanced Normalization Tools (ANTs)[94], with mutual information as the cost function for optimization. In addition, each individual T1-weighted image was nonlinearly registered to the MNI152 standard space. Both registration steps yielded a combination of affine and symmetric image normalization (SyN) transforms, which were subsequently applied or inverted, as appropriate, in later preprocessing and parcellation steps.

**MRSI partial volume correction.** To account for partial volume effects (PVE) arising from the limited spatial resolution of MRSI (5 mm isotropic), we applied a correction combining region-based voxel-wise (RBV) modeling[95] with point spread function (PSF) deconvolution to account for signal blurring. High-resolution T1-weighted structural images (1 mm isotropic) were first segmented into probabilistic tissue maps (gray matter, white matter, and CSF) using the Computational Anatomy Toolbox (CAT12)[96] for SPM12[97]. These tissue maps were then transformed into MRSI space using the inverse of the composite affine and symmetric image normalization transforms previously estimated during MRSI registration (see Section MRSI Registration). Partial volume correction was performed using the `petpvc` command-line tool[98], applying the RBV method with an isotropic Gaussian PSF of 5 mm full-width at half-maximum in all directions. Sample MRSI volume from the Geneva study, shown before and after partial volume correction, is presented in Supplementary Fig. 9.

**MRSI parcellation.** The multi-scale cortical parcellation proposed by Cammoun et al.[78] was selected for its hierarchical organization, which allows for flexible regional granularity. This parcellation provides coarse-scale segmentation with fewer but heterogeneously sized regions (scale 1), as well as finer-scale subdivisions with a greater number of parcels of relatively uniform size (scale 5). Due to the limited spatial resolution of ${}^1$H-MRSI data (5 mm isotropic) compared to the higher resolution of T1-weighted anatomical images (1 mm), scale 3 was chosen as it offers a balance between anatomical specificity and voxel-wise compatibility. This scale was integrated into the Chimera parcellation framework (see Anatomical), resulting in the LFMIHIFS-3 schema.

To project the Cammoun parcellation to MRSI space, the parcellation image in individual T1 space was transformed using the inverse of the composite affine and symmetric normalization (SyN) transforms previously estimated during MRSI registration (see Section MRSI Registration). Nearest-neighbor interpolation was used to preserve discrete parcel labels. This procedure was repeated for each subject, resulting in personalized MRSI-space parcellations and yielding five MRSI volumes (tNAA, tCr, Cho, Ins, and Glx) distributed across 277 brain regions.

In addition to the Cammoun atlas, we evaluated two functionally derived cortical parcellations: the Schaefer-200 atlas[99] and MIST197[100]. Both were integrated into the Chimera LFMIHIFS-3 schema by replacing the Lausanne (Cammoun) parcellation, resulting in the SFMIHIFS-200 and MFMIHIFS-197 schemes, respectively. These atlases were selected for their comparable number of cortical parcels and their derivation from functional connectivity analyses, offering a meaningful contrast to the anatomically driven Cammoun atlas. To assess the uniformity of these schemes, we computed the Gini coefficient of parcel sizes. All three atlases exhibited similarly low Gini values (Lausanne: 0.20, Schaefer-200: 0.21, MIST197: 0.22). To provide a structurally and functionally agnostic reference, we also implemented two geometric parcellation approaches based on regular cubic volumes. A standard MNI152 template (1 mm isotropic) was divided into cubes of 10 mm and 15 mm edge lengths, corresponding to approximately 4 and 9 MRSI voxels per parcel, respectively. These parcellations were then transformed into each subject's MRSI space and used to assess the influence of different parcellation atlases on the results presented in this study.

**Estimation of metabolic profiles.** After parcellation, regions smaller than 8 MRSI voxels (<1 cm$^3$) or those with poor MRSI coverage in more than 70% of subjects were excluded from the 277 regions, resulting in $N = 210$ retained regions. The discarded regions were mainly located in orbitofrontal and baso-temporal regions where ${}^1$H-MRSI has limited coverage due to susceptibility artifacts. These factors reduce signal-to-noise ratio and introduce spectral contamination, hindering reliable metabolite quantification in these areas. Additionally, the ${}^1$H-MRSI sequence covered only the superior portion of the cerebellum. The complete ${}^1$H-MRSI mask coverage is shown in the Supplementary Information Fig. 1. Median metabolite values were calculated inside a given parcel and z-score normalized across regions, yielding a 5-feature metabolic profile per region. To address variability in the metabolites quantification, we followed the Monte Carlo uncertainty propagation methodology described by Instrella et. al[101], repeating the process $K_{pert} = 50$ times to generate perturbed versions of the original metabolic profile. Specifically, for each voxel $(i, j, k)$, the metabolite signal was resampled from a normal distribution with mean equal to the original MRSI signal $Y_{ijk}$ and standard deviation equal to $3\sigma_{ijk}^2$, where $\sigma_{ijk}^2$ is variance corresponding to the Cramér-Rao lower bound (CRLB; see Methods MRSI reconstruction) derived from the LCModel. Mathematically,

$$Y'_{ijk} \sim \mathcal{N}\left(Y_{ijk}, 3\sigma_{ijk}^2\right) \tag{1}$$

where $Y_{ijk}$ is the original MRSI signal value. This augmentation increased the sample size of the metabolic profiles by a factor of $K_{pert}$, providing $K_{pert} - 1$ additional perturbed profiles for each brain region, in addition to the original profile.

**Metabolic similarity mode.** To visualize the spatial embedding of the $210 \times 210$ dimensional group-averaged MetSiM (Geneva sample, $N = 51$), we reduced each of its 210 high-dimensional feature vectors $\mathbf{x}_i$—corresponding to the parcel connectivity profiles—into a lower-dimensional representation. First, Principal Component Analysis (PCA) was applied for denoising and variance capture, resulting in a 50-

dimensional embedding. Subsequently, t-distributed Stochastic Neighbor Embedding (t-SNE) was employed to further reduce the dimensionality to obtain the metabolic similarity mode (MS mode) $\boldsymbol{\mu}_i$ at every nodes $i$. As prescribed by ref.[102], PCA initialization was used to stabilize and improve the convergence of t-SNE, which in turn was chosen to preserve the observed local connectivity patterns of the MetSiM.

t-SNE converts high-dimensional distances between data points into conditional probabilities (or similarities) and then maps them into a low-dimensional space. A crucial parameter in t-SNE is perplexity, which can be interpreted as a measure of how many close neighbors each point has. It also balances local and global connectivity patterns, where lower perplexity emphasizes local structures at the expense of global structure, and higher perplexity does the opposite. The method adjusts the low-dimensional output features $\boldsymbol{\mu}_i$ by minimizing the Kullback-Leibler (KL) divergence between the probability distributions in the high-dimensional and low-dimensional spaces, as computed via a t-Student-based metric.

We explored perplexity values from 20 to 60 and observed minimal changes in the KL divergence (ranging approximately from 0.55 to 0.42). Consequently, following[102], we chose a perplexity of 30 to strike a balance between preserving both local and global structures. Finally, an early exaggeration factor of 12, as suggested by[103], was applied to enhance cluster separation during the initial stages of optimization. From these parameters, we reconstructed a filtered version of the original MetSiM, whose elements $q_{ij}$ were computed using the t-Student-based metric:

$$q_{ij} = \frac{\left(1 + \| \boldsymbol{\mu}_i - \boldsymbol{\mu}_j \|^2\right)^{-1}}{\sum_{k \neq l}\left(1 + \| \boldsymbol{\mu}_k - \boldsymbol{\mu}_l \|^2\right)^{-1}} \quad (2)$$

where $\mu_i$ is the dimension-reduced representation of the $i$-th node's connectivity profile $x_i$. We then rescaled these values to $2q_{ij} - 1$ to restore the original MetSiM value range $[-1, 1]$ and verify the overall pattern conservation of the original MetSiM shown in Supplementary Fig. 2.

An intrinsic dimensionality analysis based on ISOMAP, which estimates geodesic distances on a neighborhood graph[104], revealed that the first component accounted for 53% of the variance, the second for 20%, and the third for 4%, consistently across a wide range of ISOMAP neighborhood sizes (3-20).

**Metabolic principal path construction.** In diffusion MRI tractography white matter streamlines are traced by aligning their tangent vector to the principal eigenvectors field of the diffusion tensor field, weighted by regions of strong fractional anisotropy and thereby structurally connecting distant gray matter nodes. Here, we define metabolic paths by aligning their tangent vector with the MS gradient vector field and privileging those that globally capture the overall variability of metabolic profiles.

Instead of identifying a continuous curve in space, the path we aim to construct is a sequence of edges in a discrete lattice (finite set of nodes) within the metabolic network. We proceed to introduce the definitions for a path, the MS gradient, and normalized entropy.

Let $G = (V, E, w)$ be a weighted network where $V$ is a finite set of nodes, $E \subseteq V \times V$ is a set of edges and $w: E \mapsto [-1, 1]$ is the weight function which assigns a MetSiM value $w_{ij}$ to nodes $i$ and $j$. We define a path $\gamma$ of length $L$ as an ordered sequence of nodes

$$\gamma = (l_0, l_1, \ldots, l_L),$$

such that for each integer $t (0 \leq t < L)$, the consecutive pair $(l_t, l_{t+1})$ forms an edge in the network. We designate $l_0$ as the start node and $l_L$ as the end node. We define the MS gradient between node $i$ and $j$ as the finite-

difference directional derivative of the MS mode scalar field along the edge connecting them:

$$(\boldsymbol{\nabla}\boldsymbol{\mu})_{ij} = \frac{\mu_j - \mu_i}{\| \mathbf{r}_j - \mathbf{r}_i \|} \mathbf{e}_{ij}, \quad (3)$$

where $\mu_i$ is the MS mode value at node $i$, and $d_{ij} = \|\mathbf{r}_j - \mathbf{r}_i\|$ is the Euclidean distance between nodes $i$ and $j$ and $\mathbf{e}_{ij}$ the unit vetor between node $i$ and $j$. To fully capture the overall smoothness by accumulating local fluctuations of MS mode values, we summed the $L^2$ norm of each sequential gradient ((3)) along the path to yield the total MS gradient generated along the path $\gamma$ as

$$\mathscr{G}_\mu(\gamma) = \frac{1}{N-1} \sum_{i \in E(\gamma)} \| \boldsymbol{\nabla}\boldsymbol{\mu} \|_{i, i+1} \quad (4)$$

then we write the normalized MS gradient as

$$\widetilde{\mathscr{G}}_\mu(\gamma) = \frac{\mathscr{G}_\mu(\gamma)}{\mathscr{G}_\mu(\gamma_{\text{rand}})} \quad (5)$$

where the denominator is the maximum MS gradient generated by a uniformly random MS mode sequence $\gamma_{\text{rand}}$ of length $N$ with the same start and end nodes as $\gamma$, resulting in the normalized final quantity.

We define the normalized entropy of the MS mode values along the path $\gamma = (l_0, \ldots, l_L)$ by

$$\widetilde{H}(\gamma) = -\frac{1}{H_{max}} \sum_{i=1}^{N_{\text{bins}}} p_i \log(p_i), \quad (6)$$

where is $H_{max} = \log(N_{bins})$ is the maximum entropy (entropy of a random uniform distribution). We used the Rice rule for determining the number of bins $N_{\text{bins}} = \lceil 2(L+1)^{1/3} \rceil$ of the MS mode histogram $p = (p_1, \ldots, p_{N_{\text{bins}}})\{\mu_{l_0}, \ldots, \mu_{l_L}\}$ along $\gamma$.

The optimal path should simultaneously traverse nodes which

- **Condition 1** are highly metabolically similar to their proximal neighbors (i.e., minimize MS gradient)
- **Condition 2** globally capture the highest diversity of metabolic profiles (i.e., maximize the Shannon entropy of the MS mode distribution they generate).

The first condition is analogous to optimal alignment of the tangent vector of a smooth curve in continuous space with a vector field (MS gradient), while the second condition prioritizes nodes that globally capture the principal variability of the network. Then, for a given path $\gamma = (l_0, \ldots, l_L)$, its mean total cost is

$$\mathscr{L}(\gamma) = \frac{\widetilde{\mathscr{G}}_\mu(\gamma)}{\widetilde{H}(\gamma)} \quad (7)$$

where the optimal path $\gamma^*$ minimizes:

$$\gamma^* = \arg\min_\gamma \mathscr{L}(\gamma) \quad (8)$$

Due to the non-convexity of the cost function, we employed a brute-force algorithm via Depth-First Search backtracking to:

1. identify all possible paths connecting the start and end nodes,
2. calculate the cost $C(\gamma)$ induced by each path,
3. select the paths which yielded the minimal cost value.

Given the hemispheric symmetry of the MS mode map, path-finding was performed independently in each hemisphere. In both cases, the isthmus cingulate; lowest index) served as the starting node, and the rostral anterior cingulate cortex (rACC; highest index) as the ending node. Because subcortical regions form a separate gray matter

network, we applied the same pathfinding approach to these regions by splitting each hemisphere while keeping the brainstem as a shared structure.

Results for both hemispheres–showing the total costs of the cortical and subcortical-region paths–are displayed in Supplementary Fig. 6. The optimal path $\gamma^*$ was selected from 180304 and 229670 possible pathways in the left and right neocortical hemispheres, respectively, and from 20 and 220 pathways in the left and right subcortical hemispheres. Paths with implausible trajectories, such as loops or abrupt detours, were excluded from the selection. We applied the same procedure to the other datasets, including the Geneva Study average connectivity matrix, the cytoarchitectonic similarity matrices (Von Economo and BigBrain), and the genetic co-expression matrix, with results shown in Supplementary Fig. 7. Full results are reported in Supplementary Table 3.

## Datasets
### Cytoarchitecture
**Cognitive consilience classification.** We acquired three distinct cytoarchitectonic datasets, referred to as the cortical-laminar, Von Economo, and BigBrain cytoarchitectures. The first cytoarchitecture is based on an independent modular decomposition representing the five classic cortical laminar patterns described by Von Economo and Koskinas[105]. Following the approach of Seidlitz et al.[35], we manually assigned each node from our 308-region parcellation to one of these five cortical classes, guided by the methodology outlined by Solari and Stoner[41]. The cingulate and insular cortices were assigned respectively to a sixth and a seventh class. In addition, we categorized subcortical regions–including the thalamus, hypothalamus, amygdala, hippocampus, and brainstem–into a distinct eighth class to account for their unique cytoarchitectonic characteristics. Finally, the cerebellum parcels were assigned to a ninth class, as they are composed of distinctive cell types (e.g., Purkinje cells) found exclusively in this region of the brain.

**Von Economo similarity matrix.** The Von Economo cytoarchitecture[106] represents a digitized version of the original cytoarchitectonic brain profiles described by Von Economo and Koskinas[105], forming a comprehensive brain atlas. This atlas comprises 48 'most important' distinct cortical areas and provides detailed layer-specific histological information, including neuronal count, neuron size, and cortical thickness. The Von Economo cytoarchitectonic similarity matrix was constructed as a pairwise Pearson correlation matrix of all 48 profiles.

**Big brain similarity matrix.** The BigBrain cytoarchitectonic profiles[42] were derived from the ultrahigh-resolution BigBrain dataset[43] (see Methods Big Brain Similarity Matrix), where cortical profiles capturing the laminar cell number and density of the cortex were extracted. The resulting BigBrain similarity matrix, including pairwise similarity scores between $N$ brain regions (excluding subcortical and cerebellar regions), was directly shared with us by the authors of Wei et al.[42].

**Genetic co-expression.** Gene expression data were obtained from the publicly available dataset provided by the Allen Institute for Brain Sciences (AIBS). To reduce redundancy in complementary RNA (cRNA) hybridization probes measuring overlapping gene expression, expression values were averaged across probes targeting the same gene, while probes without matched genes were excluded. This process resulted in a dataset comprising 20,152 genes across 3,702 samples.

Given the symmetry in gene expression observed between hemispheres[107], AIBS data include samples from both hemispheres for only two subjects. However, due to the under-sampling of the right hemisphere, all analyses were restricted to the left hemisphere ($n = 152$ regions). Gene expression data for each subject were mapped onto the fsaverage (MNI152) volumetric template. This was achieved by assigning samples to the nearest centroid within the left hemisphere regions

using T1-weighted MRI scans from each AIBS subject. For the two subjects with right hemisphere data, coordinates of the right hemisphere samples were reflected before mapping to ensure consistency.

Median gene expression values were calculated for each region across participants ($N = 6$) and standardized using z-scoring, resulting in a 152 × 20,152 matrix representing genome-wide expression across 152 left hemisphere regions. A 152 × 152 gene co-expression matrix was then generated by computing pairwise Pearson correlations of gene expression between each region. This co-expression matrix, along with regional expression values, was used for comparisons with the left hemisphere data of the group MetSiM. Gene ontology enrichment analysis was performed using GOATools[108].

### Statistical analysis
**MetSiM variablity.** The variability of a set of metabolic similarity matrices was assessed by calculating the mean and standard deviation of variances across the set, and then performs a t-test to determine if the observed dispersion is significantly different from zero.

The variability of a set of metabolic similarity matrices with respect to the group average was determined by calculating the Pearson correlation coefficients between each matrix and the group average matrix. The resulting $p$-values were adjusted for multiple comparisons using the Benjamini−Hochberg method, yielding the average and standard deviation of the Fisher Z-transformed correlations, along with an overall p-value.

**Spatial null models.** To assess the extent to which the observed MetSiM structure could be explained by spatial autocorrelation alone, we implemented a family of distance-based random geometric null models (RandGeom). These models generate synthetic similarity matrices in which pairwise weights depend solely on the Euclidean distance between brain regions, while reproducing key statistical features of the empirical data.

**RandGeom (full brain).** In the baseline RandGeom model, each pairwise MetSiM weight was sampled from a Gaussian distribution whose mean linearly decreased with inter-regional Euclidean distance, based on a linear regression fit to the empirical data. Formally, for any pair of brain regions $i$ and $j$, the expected weight $\mu_{ij}$ was defined as: $\mu_{ij} = a \cdot d_{ij} + b$ where $d_{ij}$ is the Euclidean distance between the centroids of regions $i$ and $j$, and $a$ and $b$ are regression coefficients estimated from the empirical MetSiM. To account for heteroscedasticity, we stratified distances into 2 mm bins and computed the residual variance within each bin. The variance $\sigma_{ij}^2$ of each pairwise connection was then sampled from the distribution of residuals corresponding to its distance bin $M_{ij}^{\text{rand}} \sim \mathcal{N}(\mu_{ij}, \sigma_{ij}^2)$.

**RandGeom (within hemisphere).** To test whether hemispheric separation influenced model fit, we constrained the RandGeom model to within-hemisphere connections only. Separate synthetic matrices were generated for left and right hemispheres using the same procedure as described above, applied independently within each hemisphere.

**RandGeom · GMAdj.** To further impose a biologically plausible spatial constraint, we constructed a variant of the RandGeom model in which only region pairs that were adjacent in the cortical gray matter were eligible for sampling. Cortical adjacency was defined based on voxel-wise Euclidean proximity of gray matter masks in MNI space. Pairwise weights were then sampled from the same distance-based distributions as in the full RandGeom model, but applied only to gray matter-adjacent region pairs.

**Adjacent parcel permutation.** To assess statistical significance when comparing spatially embedded datasets (e.g., similarity matrices, brain

region classification scores) while preserving local spatial properties, we employed a customized spatial permutation method termed AdjPerm. This approach maintains local topology by restricting permutations to occur only between adjacent regions. Specifically, we defined adjacency based on anatomical proximity, considering each brain parcel to be adjacent to its immediate neighboring parcels. During each permutation, data values were randomly shuffled among these adjacent parcels, preserving local spatial dependencies. We fixed the number of unique permutations to 1000. By maintaining local topology, this method effectively accounts for inherent spatial autocorrelation without introducing anatomically implausible configurations.

**Normalized mutual information score.** We evaluated the agreement between two 3D scalar maps using the Normalized Mutual Information score. If both datasets were spatially embedded, we supplemented the significance of the overlap score with the AdjPerm test (see Methods Adjacent Parcel Permutation).

**Inter-class overlap score.** We evaluated the agreement between two labeled datasets using the Hungarian algorithm (Kuhn-Munkres algorithm). This approach identifies an optimal one-to-one mapping between the sets of labels, maximizing the number of samples that are consistently classified under this mapping. We supplemented the significance of the overlap score with the AdjPerm test (see Methods Adjacent Parcel Permutation).

### Network based statistics
**Weighted matrix binarization.** Network graphs were generated by binarizing weighted matrices (similarity or connectivity) based on a specified edge density $\rho$. For matrices containing negative values, negative weights were rectified to their absolute values before applying the binarization threshold. This threshold was chosen to achieve the desired proportion of binarized connections $\rho$. Graph analyses were conducted across a wide range of edge densities (1 to 30%) for each weighted matrix.

**Rich-Club.** We computed the rich-club coefficient $\phi(k)$ for a given binarized network as

$$\phi(k) = \frac{2E_{>k}}{N_{>k}(N_{>k} - 1)} \quad (9)$$

where $E_{>k}$ is the number of edges among nodes with a degree greater than $k$, and $N_{>k}$ is the number of such nodes.

**Degree centrality.** We calculated the centrality of a node using degree centrality, which measures the number of direct connections a node has to other nodes in the network. For a node $i$, the degree centrality $C_D(i)$ is defined as:

$$C_D(i) = \deg(i) \quad (10)$$

where $\deg(i)$ represents the number of edges connected to $i$. Degree centrality highlights the immediate influence of a node based on its local connectivity.

**Random networks.** For each binarized graph, we assessed the statistical significance of the observed network metrics by generating an ensemble of 1000 randomized networks using the Maslov-Sneppen rewiring procedure, which preserves the number of nodes, total number of edges, and the degree distribution of the original graph while randomizing the specific wiring of edges. We then computed empirical $p$-values for each topological metric by comparing the observed value to the distribution of corresponding values across the 1000 random graphs. Specifically, for each metric, the $p$-value was

calculated as the fraction of randomized networks in which the metric was greater than or equal to the observed value.

**Higher order connectivity.** The higher-order connectivity of a weighted or binarized network matrix **A** was computed as the $k$-th order cosine similarity of **A** with itself:

$$(A^m)_{ij} = \sum_{k_1, \ldots, k_{m-1}} A_{i,k_1} \cdot A_{k_1,k_2} \cdot \ldots \cdot A_{k_{m-2},k_{m-1}} \cdot A_{k_{m-1},j} \quad (11)$$

where $\sum_{l,m} A_{k,l} \cdot A_{m,n}$ denotes the cosine product of the $l$-th node feature with the $m$-th node feature of **A**.

**Network communicability model.** To compute the influence of distant nodes from a structural connectivity matrix **S** on the state of the MetSiM **M**, we relied on a communicabilty model[109] and estimated the Green's function of the network heat equation, which corresponds to the communicability expression:

$$\mathbf{M} = \exp(g \cdot \mathbf{S}), \quad (12)$$

The decay parameter $g$ controls the influence of distant nodes structural edges on the metabolic connectivity and was estimated by minimizing the mean squared loss between the model's output (**M**) and the observed metabolic matrix (**M**$_{obs}$). Predictive performance was then quantified as the mean Spearman correlation between the predicted MetSiM (**M**) and the observed MetSiM (**M**$_{obs}$) given the optimal $g$.

**Absolute nodal correlation.** When comparing the nodal similarity between two matrices **A** and **B** of $N$ nodes, we define the absolute nodal correlation of node $i$ as the absolute value of the Pearson correlation between its connectivity profiles in **A** and **B**. Here, the connectivity profile of a matrix **X** is given by

$$\mathbf{x}_i = \{x_{ij}\}_{j \in \{1, \ldots, N\}},$$

which is the row vector containing all similarity measures of node $i$ with the other nodes in the network.

### Visualization software
Brain visualizations were generated programmatically in Python using Nilearn[110] and FURY/DIPY[111]; backgrounds were the MNI ICBM152 2009a template[112] and FreeSurfer *fsaverage* surfaces[113]. All software/templates permit inclusion of rendered images in commercial publications with proper attribution.

### Reporting summary
Further information on research design is available in the Nature Portfolio Reporting Summary linked to this article.

## Data availability
The processed derivative data (parcellation masks, within subject and group-level metabolic similarity matrices, figure source data, and analysis scripts) are available at https://github.com/MRSI-Psychosis-UP/MRSI-Metabolic-Connectome. Source data underlying the main figures are provided with this paper in the accompanying Source Data files. The full tabulations underlying the statistical analyses are provided as Supplementary Data.

## Code availability
The analysis code used in this study is available at GitHub: https://github.com/MRSI-Psychosis-UP/MRSI-Metabolic-Connectome. The exact version used for the results reported here is archived on Zenodo (https://doi.org/10.5281/zenodo.17293201) and should be cited as[114].

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

## Acknowledgements

This project was supported by the Swiss National Science Foundation (grant number 215728); P.K. was supported by a fellowship from the Adrian & Simone Frutiger foundation. We gratefully acknowledge Jacob Seidliz for providing the digitized version of cytoarchitectonic classes based on the Cognitive-Consilience dataset, and Martijn van den Heuvel and Yongbin Wei for providing the Big Brain cytoarchitectonic similarity matrix. Moreover, we gratefully acknowledge Jean-Baptiste Ledoux for the acquisition of MRI data from the Lausanne study, and Arnaud Merglen and Camille Marie Piguet for sharing the MRI dataset of the Geneva study.

## Author contributions

F.L. developed the methodology, performed the data analysis, conducted the experiments, contributed to data interpretation, and drafted the initial manuscript. P.S., and P.H. provided critical revisions. A.K. developed the MRSI technique and provided critical revisions. E.C., Y.A.-G. performed part of the data preprocessing and provided critical revisions. P.K. conceptualized the study, supervised the project, and provided critical revisions.

## Competing interests

Antoine Klauser is employed by Siemens Healthcare as a research scientist. He contributes to on-site research and development initiatives. He does not hold fiduciary responsibilities for Siemens Healthcare. All the other authors declare no competing interests.
