## [Transparent Peer Review file · Nature Communications]

Constructing the Human Brain Metabolic Connectome with MR Spectroscopic Imaging Reveals Cerebral Biochemical Organization

Corresponding Author: Professor Paul Klauser

Version 0:

Reviewer comments:

Reviewer #1

(Remarks to the Author)

- (1) This study generates a connectome based on whole brain MRSI data. Connectome has been reported with diffusion imaging data, and this study serves as the first report that uses MRSI data. The sample size is fair, and the study also includes a validation dataset. However, the use of a larger sample size is desired. The authors can consider a multi-institution study.
- (2) While this study adds the knowledge of correlation of structural connectivity and cytoarchitectonic/ genetic co-expression patterns with metabolic connectivity, the value of metabolic connectivity will become more evident when the info alters in a diseased state so as to become a disease marker.
- (3) What is the acquisition time of MRSI? Please describe the reasons for the set-up of major MRSI scan parameters, such as TE and voxel size, and pros and cons of each. How consistent are the authors' metabolic profiles with the literature values? The slice thickness may limit the evaluation of cortical structures, not only subcortical structures. This questions validity of the results.
- (4) What is the time frame to be considered as "recent" wrt "recent psychotherapy, recent use of psychotropic medications"?
- (5) The brains in a1 of Fig 1 are upside down and do not match the others.
- (6) Please correct missing words and punctuation errors in the Methods section.

(Remarks on code availability)

Reviewer #2

(Remarks to the Author)

Thank you for the opportunity to review "Constructing the Human Brain Metabolic Connectome Using MR Spectroscopic Imaging: Insights into Biochemical Organization, Cytoarchitectonic Similarity and Gene Co-expression Networks" by Lucchetti and colleagues. The authors use whole-brain spectroscopy to derive maps of 5 metabolites. They then perform inter-regional correlations and investigate the organization of this network. The data are interesting, but the methodology is convoluted and I have doubts about the biological significance of some of the findings.

1. The Introduction is more about the evolution of MRI imaging technology than about actual biological features.

2. On a related note, it is bizarre that the authors don't at least mention FDG-PET, particularly since the first human whole-brain "connectome" was actually derived using FDG. In general, that field is surprisingly given very little due regard in this manuscript.

Horwitz, B., Duara, R., & Rapoport, S. I. (1984). Intercorrelations of glucose metabolic rates between brain regions: application to healthy males in a state of reduced sensory input. *Journal of Cerebral Blood Flow & Metabolism*, 4(4), 484-499.

3. “metabolic similarity networks display distinct topological features, notably a smoothly varying gradient delineating functionally and spatially distinct yet integrated brain regions via connector hubs”. This is presented as a major finding throughout the paper, but nobody has ever found a “gradient” (low-dimensional projection of brain imaging data) that was not smooth. They are all smooth because of spatial autocorrelation induced by image smoothing.
4. On a related note, the colormap in Fig. 3 makes the brain maps very difficult to read. The quantity (which is not labeled) is a unidimensional vector – it does not need to go through the full spectrum of colors.
5. What types metabolic processes are these networks related to?
6. What do the maps for tNAA, tCr, Glx, Cho, Ins actually look like?
7. The “perturbation” procedure described in 9.2.5 is obviously important to the data quality, but I found it hard to understand whether this is a commonly used procedure or whether it was developed for the present manuscript. Can the authors demonstrate that this method correctly augments the missing data?
8. Is there any index of SNR for this type of data?
9. There is an unfortunate tendency for the authors to refer to metabolic similarity as “metabolic fibres” which is dangerously misleading because it does not refer to a physical entity but rather to a correlation coefficient derived from N=5 datapoints.
10. “We hypothesized that metabolic similarity between nodes could depend on indirect structural connections via intermediate nodes” Why would you hypothesize that? Isn't the more natural hypothesis that metabolic similarity depends on direct structural connections?
11. If I understood correctly, in final two subsections (2.4.2) the authors want to show that metabolic similarity is greater among regions with similar cytoarchitecture and gene expression. The test that goes along with this (AdjPerm, described in 9.4.2) does not seem quite right because it only permutes nodal values locally. If the authors were simply to regress out Euclidean distance from MeSIM and cyto-/gene-similarity and then correlate the residuals, are they still significantly correlated?
12. The analysis of rich clubs of the binarized MeSIM matrices is counter-intuitive. The rich club phenomenon is specifically a feature of structural connectivity, with the idea that a small number of physical hubs are more likely to be connected with each other. Why not derive the RC map for the SC network and then ask how the MeSIM is related to this architectural arrangement (similar to what the authors did in Fig. 5)?

(Remarks on code availability)

Reviewer #3

(Remarks to the Author)

Review Constructing the Human Brain Metabolic Connectome Using MR Spectroscopic Imaging.

Lucechetti et al have provided a report of an extensive amount of work in trying to create a Human Brain Metabolic Connectome from 3D magnetic resonance spectroscopic imaging data. They have employed techniques usually used in other connectome type studies, mostly from graph theory (I believe) and have developed metabolic similarity matrices (MeSiMs) for already existing cortical parcellations that have been registered to the MRSI data. In doing so they show: an organisation of metabolite profiles that suggest proximal regions are more metabolically similar to each other than they are to distal ones; that a metabolic gradient appears to exist, which they have used to develop “metabolic fibre maps”; that regions demonstrate “hub” like structure that matches to some metrics of centrality, but not to others; and that to some extent regional gene expression similarity matrices show a relationship to the metabolite similarity matrices/profiles.

This was an impressive piece of work – but I am concerned that the methodology used has driven the results seen – particularly the “metabolic fibres” work – with the choices being made forcing the results to come out as they have. I will admit upfront, my expertise is limited when it comes to the techniques and methods applied in these large scale network connectome style studies, but I followed enough to have concerns.

An example of how the choices made in the processing and analysis steps affect the final results is seen here:

Results page, 11

“Fig. 4 Metabolic Network Topology a, Binarized Group MeSiM (18% edge density) from the Geneva study (left) and associated rich club network (right). Rich-club nodes are displayed at their anatomical locations, color-coded by their MSI, with node radii scaled according to degree. Interhemispheric edges are omitted for visualization purposes. b, Core topology of the metabolic connectome represented by connector hub situated along the metabolic fibers depicting regions of metabolic similarity through a smooth gradient color-coded by the MSI. Rich-club nodes (transparent) and edges are also highlighted. c, Schematic representation of the main metabolic fibers per hemisphere running through the cortex of four brain lobes and from subcortical structures to the brainstem and cerebellum, following the rostro-caudal axis. Connector hubs are highlighted along the curves. Metabolic similarity (but interrupted connection) between the subcortical fiber and the cortical

fiber is indicated by a dashed curve. Precuneus (PCN), Pericalcarine cortex (PeriCal), Lingual gyrus (LG), Superior parietal lobule (SPL), Precentral gyrus (PreC), Lateral occipital gyrus (LOG), Inferior occipital gyrus (IOG), Inferior parietal lobule (IPL), Middle temporal gyrus (MTG), Pars triangularis (PTri), Rostral anterior cingulate cortex (rACC), Anterior insular cortex (ains), Caudate nucleus (Caud), Thalamus (Thal), Brainstem (BS), Hippocampus (Hipp), and Cerebellum (Cer)”

The schematic for figure 4.c and figure note implies that there is a functionally organised network that runs through the cortex. However, it is likely that the schematic is simply a reflection of the chosen parcellation. Due to sampling bias, nodes will naturally occur through the cortex, as only grey matter has been parcellated. Furthermore, the term “connector hub” would suggest that the identified regions are part of a larger metabolic network running through the cortex, through which projections from each “connector hub” is bridged with another. However, the DTI results would suggest that this is not the case. The researchers should 1. Acknowledge the sampling bias driving the selection of nodes in the grey matter, 2. Make it clear that there is little evidence to actually suggest these are connected hubs from this study.

There are also methodologic flaws that could severely impact the results and conclusions being drawn.

My first concern here is, “How did the authors correct for the partial volume effect in their voxels?”

That is how did they correct for the fact that given the coarser resolution of the MRSI map to the anatomical parcellations, the grey matter (GM) and white matter (WM) composition of different regions will likely differ – such that within the MRSI data one parcellation region may have 80% GM (+ 15% WM + 5 % CSF), while another more distal region the voxels may have 60% GM (+ 38% WM, + 2% CSF)? This partial volume effect is problematic on two fronts – The first is the fact that in general GM has higher concentrations of most of the metabolites being used to develop the MeSiMs, (with the exception of total Choline which may be slightly lower in GM), and the second is that the water reference used will also be impacted by this partial volume, with again, higher Water in GM compared to WM.

This is an important point, as how this is dealt with will impact all metabolite values, and could artifactually drive the metabolite similarity matrices, and metabolite gradients seen in the results, and therefore impact all the other relationships seen, from the nodal similarities, through the discussion on the metabolic fibres, to the centrality findings. The authors even suggest this is what drives the results seen:

Result page 4 “reflect metabolic dissimilarity and are likely driven by differences in tissue composition.”

Discussion, page 18

“Notably, this pattern is primarily driven by a smooth decrease in the tNAA/Cho ratio (relative to colinear levels of tCr, Glx, and Ins) from the occipital lobe (high ratio) through the parietal and frontal cortices, extending into subcortical structures, the brainstem, and the cerebellum (low ratio).

This decrease is consistent with existing MRS literature that demonstrated higher tNAA (a neuronal density marker) in gray matter–rich cortical regions but lower tNAA (and thus higher Cho, a membrane turnover marker) in white matter–rich subcortical areas such as the basal ganglia and brainstem Guan et al. (2017); Ratai et al. (2018); Zimny et al. (2013).” Other aspects in the discussion also strongly suggest the findings reported here are more related to tissue partial volume effects, and not underlying fundamental biological mechanisms, which may limit the utility of this technique as a bio-marker for other conditions. Indeed, if the results are purely being driven by tissue partial volume effects, the use of MRSI is redundant – as simple anatomical imaging would likely be more informative.

Secondly, In addition to accounting for the differing water and metabolite content due to tissue partial volume effects within the MRSI data, the authors also need to address relaxation differences between the water reference MRSI scan and the metabolite MRSI acquisition, specifically T1 related relaxation (as T2 is effectively minimised due to the ultra short TE of the FID method).

So what relaxation correction method was used to address differences in T1 relaxation between the reference water map and the metabolite map?

Did they also include the tissue partial volume effects when making this correction (for both water content and relaxation differences between grey matter (GM) and white matter (WM)?) Basically, as even with the high resolution (for MRSI) of 5 x 5 x 5 mm³ the voxels will contain a range of GM and WM content, some may be 80% GM, while others only 40% GM, and this will impact both the concentration of water present giving rise to the water signal, and how its T1 relaxation impacts that signal, and so will directly impact all metabolite concentrations as a result. See Gasparovic et al 2006 and Near et al 2020 for discussion of these points. (Gasparovic et al., 2006; Near et al., 2020)

How this is dealt with will impact all metabolite values, and could be “artifactually” driving the metabolite similarity matrices, and metabolite gradients seen in the results, and therefore impact all the other relationships seen, from the nodal similarities to the centrality findings.

Providing appropriate correction for tissue partial volume effects would actually make this study even stronger. On the one hand if the metabolite profiles, MeSiMs and metabolite gradients presented remain (qualitatively) a stronger argument for regional specificity of function and metabolism is presented. Whereas if the MeSiMs are changed as a result of now controlling for tissue partial volume, it may now support the hypothesised correlation to structural connectivity more.

I strongly feel this needs to be done before publication.

I have a few other minor concerns that I think the authors should give some thought to. The term “Metabolic fibre” does not really make any sense, except in the sense of the connectome framework they are working from. I think metabolic gradient might be a better descriptive term. Note, these gradients could simply be driven by anatomical aspects, so care should be taken as to how they are interpreted.

At the end of the manuscript I also found myself asking “So after all this very impressive technical data collection and analysis, what do we know that is new, or of some benefit?” This should be addressed better in the discussion. (once partial tissue volume effects are addressed)

Finally some minor points on the Methods:

Please use the Minimum Reporting Standards in Magnetic Resonance Spectroscopy (MRSinMRS) (Lin et al., 2021) for reporting the MRSI protocol and provide the MRSinMRS checklist. This will provide a fuller reporting of the MRSI methodology and would address the following concerns in the methods section.

MRSI

3D acquisition – list Phase encoding steps in each direction and size of CS value.

MR methods first paragraph repeats.

MRS – recommend inclusion of MRSinMRS checklist as supplement. (Lin et al., 2021).

What CS factor was used for the MRSI data? How many shots or acquisitions were averaged (Assuming 1?)

(Remarks on code availability)

I did not review the code involved, as this would be outside my main area of expertise - that of MRS analysis and interpretation.

Reviewer #4

(Remarks to the Author)

(Remarks on code availability)

Version 1:

Reviewer comments:

Reviewer #1

(Remarks to the Author)

The authors have addressed my comments satisfactorily.

(Remarks on code availability)

Reviewer #2

(Remarks to the Author)

The authors have mostly addressed my concerns and the paper is overall better. I recommend publication.

(Remarks on code availability)

Reviewer #3

(Remarks to the Author)

I have no more comments on this paper. The authors have addressed my previous concerns.

(Remarks on code availability)

Reviewer #4

(Remarks to the Author)

(Remarks on code availability)

Subject: Response Letter to Reviewers

Manuscript ID: NCOMMS-25-23028-T

Manuscript Title: Constructing the Human Brain Metabolic Connectome Using MR Spectroscopic Imaging: Insights into Biochemical Organization, Cytoarchitectonic Similarity and Gene Co-expression Networks”

,

We would like to thank you and the reviewers for your careful review, valuable feedback, and the opportunity to revise our manuscript. We have carefully considered all comments and made revisions based on the reviewers' suggestions. Given the substantial revisions required, the manuscript has undergone changes and major additions, particularly regarding the issue of spatial autocorrelation between MRSI parcels and the necessity of partial volume correction. These adjustments have affected the content of the manuscript but in no way altered the conclusions. On the contrary, they have proven that original results were not only more conclusive but also more robust than expected. We are grateful to the reviewers for raising these important points, as they have significantly improved the quality of the manuscript. Below, we provide a detailed summary of how we have addressed each concern. For ease of reference, revised sections of the manuscript are highlighted in blue, our direct responses to the reviewers are also highlighted in blue, and the specific changes made in the manuscript are highlighted in purple.

Comments Reviewer One (R1)

R1.1 : *This study generates a connectome based on whole brain MRSI data. Connectome has been reported with diffusion imaging data, and this study serves as the first report that uses MRSI data. The sample size is fair, and the study also includes a validation dataset. However, the use of a larger sample size is desired. The authors can consider a multi-institution study.*

We fully agree that expanding to larger, multi-institutional cohorts will be important to establish the generalizability of MRSI-based connectomes. In fact, concurrent studies are already underway in collaboration with two external sites, each recruiting additional participants using the same acquisition and processing pipeline. These ongoing efforts will enable us to test cross-site reproducibility and to investigate more subtle inter-individual and clinical effects. While a larger sample size would undoubtedly strengthen the findings, the present study already draws on data from two independent samples scanned in two different sites / MRI scanners, making it inherently multi-institutional. This replication in an independent sample is noted in the Introduction, where we specify that the study is based on two samples of healthy participants. Importantly, the primary goal of this work was to establish the feasibility and replicability of constructing a whole-brain connectome from MRSI data and to identify novel network features that emerge from metabolic similarity.

Added a sentence in the discussion addressing multi-institutional studies.

R1.2: *While this study adds the knowledge of correlation of structural connectivity and cytoarchitectonic/ genetic co-expression patterns with metabolic connectivity, the value of metabolic connectivity will become more evident when the info alters in a diseased state so as to become a disease marker.*

We agree that such comparisons in clinical populations are essential; however, we have deliberately placed limited emphasis on patient data in the present manuscript, as our goal here was to establish the methodological framework and demonstrate its feasibility in healthy cohorts. While follow-up studies in patients are already underway, these investigations will require additional time to complete and are beyond the scope of the current work.

No changes to the manuscript were made.

R1.3: *What is the acquisition time of MRSI? Please describe the reasons for the set-up of major MRSI scan parameters, such as TE and voxel size, and pros and cons of each. How consistent are the authors' metabolic profiles with the literature values? The slice thickness may limit the evaluation of cortical structures, not only subcortical structures. This questions validity of the results.*

Total MRSI acquisition time was 20 minutes. While this information was already reported in the Methods section, we have repeated it in the Introduction for clarity. The echo time (TE) was 1.5 ms and the voxel size was 5 mm isotropic, also as specified in the Methods. The choice of these param-

eters, as well as the trade-offs they entail, has been extensively discussed in the MRSI neuroimaging literature, particularly in relation to the specific sequence used here (3D ^1H FID-MRSI). In brief, the excite-acquire sequence scheme of the FID allows TE to be minimized to the order of the ms. Unlike conventional spin-echo MRSI, where TE is constrained by refocusing RF pulses and crusher gradients, the FID-MRSI signal acquisition starts immediately after excitation followed by a short encoding gradient (1ms), so TE is essentially the delay to the first sampling point. This approach minimizes T2 weighting therefore maximizing SNR and preserves a broad range of metabolites (five reported metabolites are resolved). The 5 mm isotropic resolution represents a balance between spatial detail, SNR, and scanning time; in combination with short TE, efficient encoding (CS) and a low-rank reconstruction, it enables whole-brain coverage without excessive SNR penalty. Detailed technical justifications for these choices are outside the scope of this paper, and we refer interested readers to the cited references in the manuscript. Our regional tNAA/tCho ratios closely mirror values reported in prior MRSI and single-voxel studies: we observe higher tNAA (relative to Cho) in cortical gray matter and elevated Cho in subcortical, glial cell-rich regions. We added a section in the Extended Data, *Validation of MRSI-derived Metabolite Ratios against Prior Literature*, where we compare our MRSI-derived metabolite levels (non- z -normalized) with previously published values. Briefly and the most crucial for our analysis is the Cho/tNAA ratio, which closely matches the values reported by Maudsley et al. (2019). Specifically, for Cho/tNAA we observed hemisphere-averaged values of ~ 0.151 (occipital), 0.170 (parietal), and 0.205 (frontal), in close agreement with Maudsley et al. (0.15, 0.16, 0.22), reproducing the expected caudal-to-rostral increase. Spatial encoding was performed with 3D phase encoding with isotropic voxel size (5 mm in all directions), so there is no “slice thickness” per se; however, partial-volume effects inevitably occur at voxel boundaries, particularly along highly folded cortical surfaces. These were accounted for in our new analyses following comments from other reviewers. To quantify sampling density, we have now included a Supplementary File (*GenevaStudy_atlas-chimeraLFMIHIFIS_scale3_npert-50_desc-voxelcount_per_parcel.csv*), which lists the number of voxels per region of interest alongside the corresponding MRSI coverage relative to the full coverage of the coregistered anatomical T1 image. The most prominent examples in the subcortical regions are: thalamic parcels, ~ 10 – 14 voxels; caudate, ~ 40 voxels; putamen, ~ 53 voxels; hippocampus, ~ 32 voxels; hypothalamic parcels, typically 0–1 voxels; and amygdala, ~ 10 voxels (56% coverage), which were excluded in both hemispheres. Finally, if the reviewer’s concern relates to the absolute values of metabolic profiles, we note that this work does not aim to interpret such absolute concentrations. Rather, our analyses are based exclusively on regional pairwise correlations of metabolite levels and focus on relative spatial patterns across the brain.

We have specified acquisition times in the Introduction, supplemented the manuscript with the minimum reporting items required by the MRSinMRS standards, and added a new Extended Data section titled *Validation of MRSI-derived metabolite ratios against prior literature*. We added a section in the Extended Data, *Validation of MRSI-derived Metabolite Ratios against Prior Literature*

R1.4: *What is the time frame to be considered as “recent” wrt “recent psychotherapy, recent use of psychotropic medications”?*

We thank the reviewer for pointing out this inaccuracy. “Recent” refers to the last 6 months in

both cases. This has been clarified in the Methods .

Precisions of the time frame were added to the corresponding paragraph (lines 1161-1163).

R1.5: *The brains in a1 of Fig 1 are upside down and do not match the others.*

We thank the reviewer for pointing out this inconsistency, which has now been corrected in the updated version of Fig. 1

Refined Fig.1 and applied correction as noted.

R1.6: *Please correct missing words and punctuation errors in the Methods section.*

We thank the reviewer for their careful reading. These unfortunate typographical errors have all been corrected.

Revised.

Comments Reviewer Two (R2)

R2.1. *The Introduction is more about the evolution of MRI imaging technology than about actual biological features.*

The authors agree with this statement and have amended the manuscript accordingly.

We have considerably shortened the section of the Introduction that described the evolution of MRI technology, and we have expanded the motivation for using MRSI as a metabolic imaging technique (lines 98-124). In addition, we have added more background on the biological relevance of the metabolites measurable with MRSI. While we acknowledge that the precise biological interpretation of certain metabolites remains, to some extent, elusive when measured in vivo, this added context better frames the rationale for our study within a biological perspective rather than a purely technological one.

R2.2. *On a related note, it is bizarre that the authors don't at least mention FDG-PET, particularly since the first human whole-brain "connectome" was actually derived using FDG. In general, that field is surprisingly given very little due regard in this manuscript. Horwitz, B., Duara, R., & Rapoport, S. I. (1984). Intercorrelations of glucose metabolic rates between brain regions: application to healthy males in a state of reduced sensory input. *Journal of Cerebral Blood Flow & Metabolism*, 4(4), 484-499.*

Indeed, we described our work as the first MRSI-based metabolic connectome, rather than claiming it as the first metabolic connectome overall. However, we agree with the reviewer that a mention of metabolic connectomes obtained from FDG-PET studies was missing. Given our focus in the Introduction on MRI-based techniques, we did not expand on PET there. The comparison between FDG-PET and MRSI metabolic imaging — including methodological differences and complementary strengths — is now addressed in the Discussion section.

We have added a new paragraph in the Discussion section (lines 1017–1032) comparing FDG-PET and MRSI metabolic imaging.

R2.3. *"metabolic similarity networks display distinct topological features, notably a smoothly varying gradient delineating functionally and spatially distinct yet integrated brain regions via connector hubs". This is presented as a major finding throughout the paper, but nobody has ever found a "gradient" (low-dimensional projection of brain imaging data) that was not smooth. They are all smooth because of spatial autocorrelation induced by image smoothing.*

We would like to thank the reviewer for pointing this out, as it not only required substantial additions to the manuscript but also helped us clarify an important source of confusion, which we must admit was not fully resolved or understood by ourselves, thereby contributing significantly to the overall quality of the work. We recognize that the term gradient has become a term of art in the neuroimaging literature, used to describe any low-dimensional projection of a higher-dimensional network matrix. This is somewhat unfortunate, since in mathematics a gradient refers to a vector

field (or a scalar field subjected to a differentiation operator, as in the case where differentiating the electric potential yields the electric field), and this discrepancy in terminology can make our claim potentially confusing. While “gradients” reported in neuroimaging are indeed partially smooth due to spatial autocorrelation, their true gradient do not necessarily vary smoothly in the same way everywhere. Features such as fibre density or cortical thickness, for example, do not evolve linearly and certainly not monotonically (i.e., strictly increasing or decreasing). This distinction is crucial, and it motivated our need to clearly differentiate between “gradients” as commonly used in the neuroimaging literature (which we originally termed the MSI, now referred to as the MS mode and where term mode occur oftentimes in gradient studies) and true mathematical gradients, defined as the local spatial variations of these modes (a distinction never previously addressed in the literature). To make our claim precise and highlight its novelty, we have revised the manuscript to clearly delineate these definitions and have renamed the key terms consistently throughout.

- Terminology change: We now refer to the “metabolic similarity index” as the metabolic similarity mode (MS mode). In line with existing literature, we use “mode” rather than “gradient” to avoid confusion with the true mathematical gradient, which we compute explicitly in our work.
- Mathematical definition: We define the metabolic similarity gradient (MS gradient) as the edge-wise spatial differentiation of the MS mode, yielding a true vector field whose direction depends on the node pair considered.
- Normalized gradient: We define the normalized MS gradient as the norm of this vector field along specific node sequences (principal paths).
- Illustration: A scheme has been added to Fig. 4 to visually distinguish between modes and gradients. We hope that it will minimize the risk of confusion for readers unfamiliar with this difference.

Given these definitions, we refined our claim from describing a “smoothly varying gradient” to a “monotonically increasing MS mode.” In the Discussion (lines 940-959), we contrast this property with other reported “gradients” that do not satisfy monotonicity with two exceptions. As we demonstrate, this pattern is not simply a by-product of spatial autocorrelation but instead emerges from the co-occurrence of local metabolic similarity and global metabolic diversity. We have also expanded our discussion of monotonicity, emphasizing that it is not an artefactual construction but rather the outcome of these two counterbalancing effects. In addition, Fig.4 now includes simulations (10,000 pure random geometric networks) illustrating why both effects must be present in the network to give rise to these paths. Finally, we added a gradient analysis to contrast this unique property of MeSiMs with structural connectivity, cytoarchitectural similarity, and genetic co-expression matrices, showing that monotonicity is not a general property of all networks (particularly the first derived from diffusion MRI, which inherently contain spatial autocorrelation) but emerges only in the genetic co-expression network, which, like the metabolic network, satisfies both local and global effects, albeit in the genetic co-expression space.

Added a mathematical definition of the MS gradient in Methods Section ABC. Included a schematic explanation of the MS mode and MS gradient in Fig. 4. Clarified all mentions of “smoothly varying” accordingly, and emphasized our main finding by specifying a “monotonically varying” MS mode, and removing any confusion between modes and gradients.

R2.4. *On a related note, the colormap in Fig. 3 makes the brain maps very difficult to read. The quantity (which is not labeled) is a unidimensional vector – it does not need to go through the full spectrum of colors.*

The colormap was chosen deliberately, as the mapped quantity (MS mode) contains substantial variance when embedded in 3D space, requiring a broader spectral range than a simple three-color scale (e.g., blue–white–red) with two contrasting colors and a single intermediate. Applying a clustering algorithm with an elbow method to determine the optimal number of distinct MS mode centroids yielded a range of 6–9 clusters, which naturally benefits from a fuller spectral colormap. For comparison, we tested a dark-to-white colormap (see Figure below): in this case, occipital regions completely lose contrast from each other, cortical regions are less distinguishable, despite being strongly metabolically anticorrelated. This illustrates that a restricted contrast range would obscure relevant distinctions in the data. We are aware that brain gradients and modes are often displayed using simpler colormaps, typically with a two-color scale when the gradient spans only two functionally distinct regions (e.g., sensory–motor or cortical–subcortical). In contrast, this is not the case here, where multiple regions are delineated along a vector that is arguably not unidimensional but more likely multidimensional, as it is driven by the covariation of five metabolite levels. Moreover, in manifold learning approaches such as the one applied here, it is common practice to represent the unfolded space using a full spectral colormap, particularly when the manifold follows a more complex and highly non-linear mode pattern, as in the rostral–caudal to subcortical progression observed in this work. The take-home message of Fig. 3 is not to inform the reader of the absolute value assigned to each brain parcel given the chosen color mapping. Rather, it is to demonstrate that the low-dimensional representation clearly reveals the two key effects discussed: (1) strong contrast between functionally and spatially distinct regions, and (2) a degree of similarity between functionally similar and spatially proximal regions. The chosen colormap was selected to balance these two aspects in a way that conveys the full visual picture. Regarding the comment “The quantity (which is not labeled)”, we are not entirely sure what the reviewer is referring to. If this is a major point of contention, we would appreciate clarification. The colormap is indeed labeled — previously as Metabolic similarity index [AU] and now as Metabolic similarity mode [AU]. Units are not critical here, as we are concerned only with the relative contrasts (i.e., metabolic dissimilarity between regions), not the absolute magnitude of the MS modes. We have added scale tick labels to other colormaps where it was appropriate.

No action taken.

R2.5. *What types metabolic processes are these networks related to?*

We were not entirely certain which network the reviewer was referring to, but we assume the question concerns the metabolic similarity network, which is built on the measures of five metabolites related to energy metabolism, excitatory neurotransmission, membrane turnover, and glial cell activity. To address this, we have added background on the biological underpinnings of the five metabolite compounds that contribute to the construction of the metabolic similarity network. Please let us know if further clarification is needed.

Added biological background on the five MRSI metabolite signals in the Introduction (lines 117-124) and further detailed their indirect role in glucose metabolism for PET/MRSI comparisons

Figure 1:

(1017–1032).

R2.6. *What do the maps for tNAA, tCr, Glx, Cho, Ins actually look like?*

We agree that these important illustrations were missing

Added axial, sagittal, and coronal slices of the 5-metabolite MRSI maps in Fig. 1. In the Supplementary Material (Fig.9), we added 8 axial slices of the same maps, supplemented with the effect of partial volume correction, for interested readers and in response to another reviewer’s comments.

R2.7. *The “perturbation” procedure described in 9.2.5 is obviously important to the data quality, but I found it hard to understand whether this is a commonly used procedure or whether it was developed for the present manuscript. Can the authors demonstrate that this method correctly augments the missing data?*

What we implement is a Monte-Carlo uncertainty propagation of our MRSI measurements, grounded in the LCMoel-derived Cramér–Rao Lower Bounds (CRLBs). Although tailored here for region-level metabolic profiles, the core methodology is well established in both MRS and especially in high-energy physics. **1. Description of the Method** Each voxel concentration Y_{ijk} is resampled according to its CRLB-based variance:

$$Y'_{ijk} \sim \mathcal{N}(Y_{ijk}, 3\sigma_{ijk}^2),$$

where σ_{ijk}^2 is the variance corresponding to the CRLB issued by the LCMoel. We repeat this K_{pert} times ($K_{\text{pert}} = 50$) and aggregate into parcel-median metabolite profiles, yielding K_{pert} plausible realizations per region. **2. Rationale as “Augmentation”**

- **Uncertainty-driven augmentation:** Analogous to adding sensor noise in computer-vision, we inject measurement-error-based noise to produce replicates that capture true quantification uncertainty.
- **Not simple imputation:** We do *not* fill missing voxels, but propagate each voxel’s fitting error through to region-level summaries.
- **Statistical robustness:** Treating these replicates as an ensemble enables bagging-style variance reduction and stability testing of our metabolic similarity matrices.

By fixing $K_{\text{pert}} = 50$ for all subsequent analyses, (manuscript section “MetSiM reliability analysis”) the inter-subject MeSiM variance is stabilized with respect to the the population average. This also makes the matrices more stable following metabolite imputation from the MeSiM correlation (Spearman between region-wise metabolic profiles) and achieving a balance between computational cost and statistical robustness. Our perturbation procedure therefore validly propagates the measured uncertainty in the spectral quantification of the metabolic profiles by creating a distribution of the most probable measures (estimated mean +- CRLB variance) and supplementing it with additional measures to the correlation output. The result is that we MetSiM construction is now demonstrably stabilized without overfitting or mis-imputing missing data. Because it is not data augmentation in the classical data science or statistics sense we left it out and replaced it by the more exact term of uncertainty propagation.

We replaced "data augmentation" by "Monte-Carlo uncertainty propagation" in the Results section (lines 162-166). We added more background to the appropriate Methods Section "10.2.7 Estimation of Metabolic Profiles".

R2.8. *Is there any index of SNR for this type of data?*

Yes, such an index exists, as briefly mentioned in the Results section, and is derived from the LCModel metabolite quantification process. We did not directly use it in the present study; however, a parallel study employed it to construct a quality mask identifying voxels of sufficient quality for further analyses and excluding those with low SNR. In our case, the construction of the quality mask for the MetSiM connectome (see Supplementary Material, *MetSiM Connectome Coverage*) was based instead on setting a threshold for the minimum number of MRSI coverage per parcel to contribute to a MetSiM weight. We further relied on the CRLBs of the five metabolite maps as the basis for our Monte Carlo uncertainty propagation procedure. For completeness, we have now supplemented the Supplementary Figures with an illustration of the SNR map together with the CRLB maps for the five metabolites (Supplementary Fig.10), for interested readers. From those images, interested reader should be able to visually appreciate the insufficient quality of the MRSI signal at the level of the inferior temporal lobe and orbito-frontal regions.

Added Supplementary Fig.10

R2.9. *There is an unfortunate tendency for the authors to refer to metabolic similarity as "metabolic fibres" which is dangerously misleading because it does not refer to a physical entity but rather to a correlation coefficient derived from $N=5$ datapoints.*

We agree that referring to "metabolic similarity" as "metabolic fibres" may cause confusion, as this does not correspond to a physical tract but rather emerges from statistical relations. In the revised manuscript, we have therefore replaced the term "metabolic fibre" with the more rigorous definition of a "metabolic principal path." By construction, this principal path represents a continuous trajectory through brain coordinates, determined by locally defined scalar values that are themselves derived from the similarity matrix. While each local similarity reflects correlations computed from five metabolite measurements, the path is not simply a correlation coefficient but rather a curve constrained by the full similarity structure. In this sense, our initial terminology was conceptually motivated by an analogy with white matter fibres in tractography, where reconstructed WM fibres correspond to locally coherent regions of high anisotropy, and where streamline propagation follows the principal diffusion direction (i.e., the local tangent vector is aligned with the principal eigenvector of the diffusion tensor). In our case the principal path traced through spatial coordinates where metabolic similarity remains coherent (does not vary). Nevertheless, to avoid any misunderstanding, we now adopt the term "metabolic principal path" (drawn from the statistics and model fitting literature) which we define formally in the Methods Section "Metabolic Principal Paths". In the current revised manuscript, and following the comments of other reviewers, we show that these paths, although they do not correspond to a biophysical tangent object, nonetheless emerge from an underlying complex topology that arises only when a certain degree of local metabolic similarity is reached concurrently with high global metabolic diversity. They therefore represent more than a mere mathematical construct and constitute a measurable signature that reflects the joint occurrence of these two biologically grounded local and global effects. We also discuss that these paths are highly specific to the MRSI connectome topology, which justifies our need to highlight them as a key finding.

We replaced all mentions of “metabolic fibre” with “metabolic principal path” and expanded the discussion (lines 942 - 996) to clarify their definition and biological significance.

R2.10. *“We hypothesized that metabolic similarity between nodes could depend on indirect structural connections via intermediate nodes” Why would you hypothesize that? Isn’t the more natural hypothesis that metabolic similarity depends on direct structural connections?*

We started testing the relationship with direct structural connections, which showed only a very low correlation ($r = 0.05$). We then extended the analysis to higher-order connections and communicability deliberately “to play devil’s advocate” and to rigorously demonstrate that the two networks are largely independent. Indeed, we had also previously shown that the correlation between structural and metabolic similarity peaks at second-order connections, suggesting that metabolically similar nodes are not typically directly connected, but rather tend to share a common structural neighbor. This is consistent with the observation that many nodes project structurally to the same hubs.

Rephrased sentence line 610-611 to improve clarity.

R2.11. *If I understood correctly, in the final two subsections (2.4.2) the authors want to show that metabolic similarity is greater among regions with similar cytoarchitecture and gene expression. The test that goes along with this (AdjPerm, described in 9.4.2) does not seem quite right because it only permutes nodal values locally. If the authors were simply to regress out Euclidean distance from MeSiM and cyto-/gene-similarity and then correlate the residuals, are they still significantly correlated?*

We thank the reviewer for raising this important point, which we acknowledge was underexplained in the original manuscript. We were aware of possible spatial autocorrelation effects and have now explicitly addressed them. In particular, the main reduction in correlation values arose from applying partial volume correction, while regressing out Euclidean distance produced only a minor additional decrease. Statistical significance was negligibly affected in both cases. For completeness, we now report correlations both before and after spatial correction, even though we consider this step unnecessary once partial volume effects and MRI-related spatial blurring have been corrected (via GM-only extraction and 5 mm point spread function deconvolution). Overall, these corrections confirm that the manuscript’s conclusions remain unchanged.

We have supplemented all MeSiM correlations with results both before and after spatial autocorrelation correction (Section “Metabolic Similarity and Cytoarchitecture” and “Metabolic Similarity and Genetic Co-Expression”).

Comments Reviewer 3 (R3)

R3.1. *The schematic for figure 4.c and figure note implies that there is a functionally organised network that runs through the cortex. However, it is likely that the schematic is simply a reflection of the chosen parcellation. Due to sampling bias, nodes will naturally occur through the cortex, as only grey matter has been parcellated. Furthermore, the term “connector hub” would suggest that the identified regions are part of a larger metabolic network running through the cortex, through which projections from each “connector hub” is bridged with another. However, the DTI results would suggest that this is not the case. The researchers should 1. Acknowledge the sampling bias driving the selection of nodes in the grey matter, 2. Make it clear that there is little evidence to actually suggest these are connected hubs from this study.*

We thank the reviewer for these constructive points. 1. We agree that using a cortical GM-based parcellation can introduce sampling bias. In the original submission we assessed robustness across multiple atlases (Schaefer and MIST). In the revision, we now state explicitly that a GM-based atlas is used during MetSiM parcellation, and we add two anatomical-/modality-agnostic cubic parcellations (10 mm and 15 mm). The corresponding MS-mode maps and similarity matrices are shown in Supplementary Fig. 3 and exhibit strong alignment with the Lausanne cortical parcellation ($MI \approx 0.99$), suggesting that our findings are not driven by atlas choice. For clarity, we also note that our use of “functional” follows connectomics convention (modules defined in GM, not WM pathways). We have added an explicit sentence acknowledging the GM sampling frame in the manuscript and are happy to expand this clarification further if not addressed appropriately. 2. We also recognize that some of our terminology may have unintentionally confused metabolic connectivity/similarity with structural connectivity. In the first draft, referring to “metabolically connected” or “metabolic structural hubs” could be misinterpreted as implying anatomical projections. We have corrected this by consistently replacing such terms with “metabolic similarity hubs,” which explicitly denote hubs defined by similarity of metabolic profiles rather than structural connections. We also refrained from using the terms “metabolically connected”.

Explicitly included GM-based atlas in the parcellation step of MetSiM construction. Added cubic parcellations (10 mm and 15 mm) with corresponding MS mode and similarity matrices in Supplementary Fig. 3. Revised terminology to consistently use “metabolic similarity hubs” and avoid implying structural connectivity.

R3.2. partial volume effect *here are also methodologic flaws that could severely impact the results and conclusions being drawn. My first concern here is, “How did the authors correct for the partial volume effect in their voxels?” That is how did they correct for the fact that given the coarser resolution of the MRSI map to the anatomical parcellations, the grey matter (GM) and white matter (WM) composition of different regions will likely differ – such that within the MRSI data one parcellation region may have 80% GM (+ 15% WM + 5 % CSF), while another more distal region the voxels may have 60% GM (+ 38% WM, + 2% CSF)? This partial volume effect is problematic on two fronts – The first is the fact that in general GM has higher concentrations of most of the metabolites being used to develop the MeSiMs, (with the exception of total Choline which may be slightly lower in GM), and the second is that the water reference used will also be*

impacted by this partial volume, with again, higher Water in GM compared to WM. This is an important point, as how this is dealt with will impact all metabolite values, and could artifactually drive the metabolite similarity matrices, and metabolite gradients seen in the results, and therefore impact all the other relationships seen, from the nodal similarities, through the discussion on the metabolic fibres, to the centrality findings. The authors even suggest this is what drives the results seen: Result page 4 “reflect metabolic dissimilarity and are likely driven by differences in tissue composition.”

Response We thank the reviewer for highlighting this crucial point. In the revised manuscript, we have now implemented a voxel-wise partial volume correction using the region-based voxel-wise (RBV) method, as described in the Methods section (“MRSI Partial Volume Correction”). This approach explicitly accounts for the varying GM, WM, and CSF contributions within each voxel and is widely used in PET studies. In addition, to further reduce the impact of spatial blurring and partial volume contamination, we applied a 5mm point spread function (PSF) deconvolution to the MRSI data. A sample of this procedure is shown in Supplementary Fig. 9, where boundary regions between GM–WM and GM–CSF are more clearly accentuated; in particular, the outline of the ventricles is better defined, and the overall images appear less smoothed. Together, these corrections mitigate the tissue-composition bias that could otherwise drive spurious similarities in metabolite profiles and ensure that our reported MeSiM gradients and network properties are not artefacts of partial volume effects. Key results were only minimally affected (the correlation of MeSiM with other networks decreased slightly), and the conclusions remain unchanged. We note that in our processing pipeline, metabolite concentrations were first referenced to the water signal and partial volume correction was then applied. We acknowledge that this ordering means that tissue-dependent differences in water content (higher in GM, lower in WM, highest in CSF) may already influence water-referenced metabolite values. However, our region-based voxel-wise correction explicitly models the GM, WM, and CSF composition of each voxel, and thereby compensates for this effect. Combined with the 5 mm PSF deconvolution, this ensures that residual biases from water scaling are minimized. We have now clarified this point in the Methods section.

Added voxel-wise partial volume correction (RBV) and 5 mm PSF deconvolution to the Methods (“10.2.5 MRSI Partial Volume Correction”) and applied these corrections to all analyses. Highlighted changes in results in blue. Added Supplementary Fig. 9, to highlight visually the effect of the PV correction.

R3.3. Discussion, page 18 tNAA/Cho *“Notably, this pattern is primarily driven by a smooth decrease in the tNAA/Cho ratio (relative to colinear levels of tCr, Glx, and Ins) from the occipital lobe (high ratio) through the parietal and frontal cortices, extending into subcortical structures, the brainstem, and the cerebellum (low ratio). This decrease is consistent with existing MRS literature that demonstrated higher tNAA (a neuronal density marker) in gray matter–rich cortical regions but lower tNAA (and thus higher Cho, a membrane turnover marker) in white matter–rich subcortical areas such as the basal ganglia and brainstem Guan et al. (2017); Ratai et al. (2018); Zimny et al. (2013).” Other aspects in the discussion also strongly suggest the findings reported here are more related to tissue partial volume effects, and not underlying fundamental biological mechanisms, which may limit the utility of this technique as a bio-marker for other conditions. Indeed, if the results are purely being driven by tissue partial volume effects, the use of MRSI is redundant – as*

simple anatomical imaging would likely be more informative.

We thank the reviewer for raising this important concern. We agree that, without correction, partial volume effects could drive apparent metabolite gradients. To address this, we have now implemented voxel-wise partial volume correction (RBV) together with 5 mm PSF deconvolution, thereby removing GM/WM/CSF mixing as a confound and extracted the only GM contribution from the signal. Importantly, after these corrections the reported decrease in the tNAA/Cho ratio persists. To avoid any ambiguity, we have revised the relevant section of the Discussion to clarify that our interpretation does not rely on GM–WM differences, but instead reflects established biological variation within GM itself. Specifically, we emphasize the higher neuronal density in cortical GM (particularly in the occipital lobe) compared to the greater glial content in subcortical GM, which exhibits increased membrane turnover. This cell-type–related distinction is consistent with prior MRS literature (we give some references) and provides a biological basis for the observed metabolic gradients that goes beyond partial volume artefacts.

Added explicit statement that PV correction (RBV + PSF deconvolution) was applied; rephrased the Discussion (lines 926 - 937) to highlight cell-type differences within GM rather than GM–WM mixing.

R3.4. Discussion, T1 relaxation corrections *Secondly, In addition to accounting for the differing water and metabolite content due to tissue partial volume effects within the MRSI data, the authors also need to address relaxation differences between the water reference MRSI scan and the metabolite MRSI acquisition, specifically T1 related relaxation (as T2 is effectively minimised due to the ultra short TE of the FID method). So what relaxation correction method was used to address differences in T1 relaxation between the reference water map and the metabolite map?*

The reviewer is correct that, given the short TR, the reconstructed metabolite maps are not absolute concentrations but are weighted by the metabolite-specific T1 relaxation times in GM and WM. A global scaling factor is also introduced by the T1 weighting of the water reference signal. However, no explicit T1 correction was applied. This decision was based on the following considerations:

- A T1 correction of both metabolites and water would act as a tissue-specific scaling factor (different for GM and WM), but, because each GM parcel is scaled uniformly, would not alter relative differences between GM parcels.
- Pairwise correlations across GM parcels—the basis of the MetSiM construction—are unaffected by multiplicative prefactors, and therefore remain unchanged by the absence of T1 correction.
- Any potential confound arising from GM–WM composition differences — leading to variations in the T1-weighting of metabolite and water signals — is already addressed by our voxel-wise partial volume correction.

For these reasons, we believe that applying T1 corrections would not affect at all the reported MetSiM correlations or the conclusions of the study.

Added a sentence in Methods Section "MRSI reconstruction" explaining the lack of necessity for applying T_1 and T_2 corrections (lines 1270-1276).

R3.5. Nomenclature: metabolic fibre *I have a few other minor concerns that I think the authors should give some thought to. The term “Metabolic fibre” does not really make any sense, except in the sense of the connectome framework they are working from. I think metabolic gradient might be a better descriptive term. Note, these gradients could simply be driven by anatomical aspects, so care should be taken as to how they are interpreted.*

We thank the reviewer for pointing this out. We agree that the term “metabolic fibre” was imprecise and have now replaced it with “principal path,” which is the correct rationale for this construction. In statistics, a principal path is a trajectory that maximizes explained variance in a high-dimensional space, thereby capturing the dominant axis of variation while preserving a continuous sequence. This more accurately reflects the methodological basis of our approach. We also refrained from adopting the term “metabolic gradient,” since no differentiation step was applied at that stage. We also recognize that, in the neuroimaging literature, “gradient” has unfortunately become a term of art, used broadly to denote low-dimensional embeddings of high-dimensional connectivity/similarity data. However, in mathematics a gradient refers specifically to a vector field (or the derivative of a scalar field, e.g., the electric field as the gradient of the potential). This discrepancy can create confusion, particularly as our work explicitly computes true mathematical gradients of the metabolic similarity mode (MS mode as the first component of the dimensionality reduction). To clarify this distinction, we have revised the manuscript as follows:

- **Terminology change:** We now refer to the “metabolic similarity index” as the “metabolic similarity mode” (MS mode). In line with existing literature, we use “mode” rather than “gradient” to avoid confusion with the true mathematical gradient.
- **Mathematical definition:** We define the “metabolic similarity gradient” (MS gradient) as the edge-wise spatial differentiation of the MS mode, yielding a true vector field whose direction depends on the node pair considered.
- **Normalized gradient:** We define the normalized MS gradient as the norm of this vector field along specific node sequences (principal paths).
- **Illustration:** We added a schematic in Fig. 4 to visually distinguish between modes and gradients, to aid readers unfamiliar with this difference.

Replaced “metabolic fibre” with “principal path.” Clarified terminology by distinguishing between “mode” and “gradient” (with explicit definitions). Added an illustration in Fig. 4 to illustrate the differences.

R3.6. Novelty *At the end of the manuscript I also found myself asking “So after all this very impressive technical data collection and analysis, what do we know that is new, or of some benefit?” This should be addressed better in the discussion. (once partial tissue volume effects are addressed)*

In the revised Discussion, we have added a synthesis of the key insights that emerge from our work after partial volume correction. Before listing these findings, it is worth emphasizing that the main novelty lies in the fact that this is, to our knowledge, the first topological analysis of such networks,

made possible only by very recent advances in whole-brain MRSI. This analysis not only replicates well-established principles of brain network organization but, by leveraging a fundamentally new type of MRI modality, provides an additional dimension of investigation beyond the purely spatial domain of structural MRI and the spatio-temporal domain of functional MRI. Specifically, we show that well-known properties of complex brain networks, such as the balance between global integration and local specialization, are recapitulated in the metabolic domain, where they can now be directly linked to regional variation and interplay of metabolites which serve as direct measure of cellular composition and function. This framework allows us to reinterpret the separation and integration of canonical functional modules (e.g., sensory, motor, limbic) through the lens of metabolic organization. Furthermore, we reinforce this statement by demonstrating that metabolic similarity is strongly shaped by cytoarchitecture and genetic co-expression, thereby justifying it as a biologically meaningful proxy for underlying cellular and molecular processes. For what is a purely exploratory study of the potential of an MRSI-based metabolic connectome, we believe these contributions are insightful, and they lay a robust foundation for future investigations into disease-related alterations in such networks.

We have expanded parts in the Discussion to better highlight the significance of these paths (lines 942-996) and concluding paragraph synthesizing the novelty and significance of our findings (lines 1045-1051).

R3.7. Minor points *Finally some minor points on the Methods: Please use the Minimum Reporting Standards in Magnetic Resonance Spectroscopy (MRSinMRS) (Lin et al., 2021) for reporting the MRSI protocol and provide the MRSinMRS checklist. This will provide a fuller reporting of the MRSI methodology and would address the following concerns in the methods section.*

- *MRSI 3D acquisition – list Phase encoding steps in each direction and size of CS value.*
- *MR methods first paragraph repeats.*
- *MRS – recommend inclusion of MRSinMRS checklist as supplement. (Lin et al., 2021).*
- *What CS factor was used for the MRSI data? How many shots or acquisitions were averaged (Assuming 1?)*

We have completed the Minimum Reporting Standards for in vivo Magnetic Resonance Spectroscopy (MRSinMRS) checklist and provided it as a CSV file in the Supplementary Materials. We now reference this file in the Methods (“MRI acquisition”). Because the two samples were acquired independently with study-specific MR scanners and MRSI parameter sets, we report them separately in the checklist rather than repeating them verbatim in the main text. A single average/measurement was acquired in both samples, and the compressed-sensing acceleration factor was $\tilde{3}.3$ (reported as 3). Pre-processing (low-rank + TGV reconstruction with lipid/water removal) and quantification (LCModel, Phase encoding steps, metabolite groupings) are fully detailed in the checklist.

Added “MRSinMRS.csv” to the Supplementary Material.